# 3DP3: 3D Scene Perception via Probabilistic Programming

**Nishad Gothoskar**[1]        **Marco Cusumano-Towner**[1]        **Ben Zinberg**[1]

**Matin Ghavamizadeh**[1]        **Falk Pollok**[2]        **Austin Garrett**[1]

**Joshua B. Tenenbaum**[1]        **Dan Gutfreund**[2]        **Vikash K. Mansinghka**[1]

[1]MIT        [2]MIT-IBM Watson AI Lab

{nishad,marcoct,bzinberg,mghavami,jbt,vkm}@mit.edu
{falk.pollok,austin.garrett}@ibm.com        dgutfre@us.ibm.com

## Abstract

We present 3DP3, a framework for inverse graphics that uses inference in a structured generative model of objects, scenes, and images. 3DP3 uses (i) voxel models to represent the 3D shape of objects, (ii) hierarchical scene graphs to decompose scenes into objects and the contacts between them, and (iii) depth image likelihoods based on real-time graphics. Given an observed RGB-D image, 3DP3's inference algorithm infers the underlying latent 3D scene, including the object poses and a parsimonious joint parametrization of these poses, using fast bottom-up pose proposals, novel involutive MCMC updates of the scene graph structure, and, optionally, neural object detectors and pose estimators. We show that 3DP3 enables scene understanding that is aware of 3D shape, occlusion, and contact structure. Our results demonstrate that 3DP3 is more accurate at 6DoF object pose estimation from real images than deep learning baselines and shows better generalization to challenging scenes with novel viewpoints, contact, and partial observability.

## 1 Introduction

A striking feature of human visual intelligence is our ability to learn representations of novel objects from a limited amount of data and then robustly percieve 3D scenes containing those objects. We can immediately generalize across large variations in viewpoint, occlusion, lighting, and clutter. How might we develop computational vision systems that can do the same?

This paper presents a generative model for 3D scene perception, called 3DP3. Object shapes are learned via probabilistic inference in a voxel occupancy model that coarsely captures 3D shape and uncertainty due to self-occlusion (Section 4). Scenes are modeled via hierarchical 3D scene graphs that can explain planar contacts between objects without forcing scenes to fit rigid structual assumptions (Section 3). Images are modeled by real-time graphics and robust likelihoods on point clouds. We cast 3D scene understanding as approximate probabilistic inference in this generative model. We develop a novel inference algorithm that combines data-driven Metropolis-Hastings kernels over object poses, involutive MCMC kernels over scene graph structure, pseudo-marginal integration over uncertain object shape, and existing deep learning object detectors and pose estimators (Section 5). This architecture leverages inference in the generative model to provide common sense constraints that fix errors made by bottom-up neural detectors. Our experiments show that 3DP3 is more accurate and robust than deep learning baselines at 6DoF pose estimation for challenging synthetic and real-world scenes (Section 6). Our model and inference algorithm are implemented in the Gen [11] probabilistic programming system.

35th Conference on Neural Information Processing Systems (NeurIPS 2021).

## 2   Related Work

**Analysis-by-synthesis approaches to computer vision** A long line of work has interpreted computer vision as the inverse problem to computer graphics [22, 39, 26, 23]. This 'analysis-by-synthesis' approach has been used for various tasks including character recognition, CAPTCHA-breaking, lane detection, object pose estimation, and human pose estimation [40, 36, 27, 30, 18, 31]. To our knowledge, our work is the first to use an analysis-by-synthesis approach to infer a hierarchical 3D object-based representation of real multi-object scenes while exploiting inductive biases about the contacts between objects.

**Hierarchical latent 3D scene representations** We use a scene graph representation [41] that is closely related to hierarchical scene graph representations in computer graphics [9]. Unlike in graphics, we address the *inverse problem* of inferring hierarchical scene graphs from observed image data. Inferring hierarchical 3D scene graphs from RGB or depth images in a probabilistic framework is relatively unexplored. One concurrent[1] and independent work, Generative Scene Graph Networks (GSGN [13]), proposes a variational autoencoder architecture for decomposing images into objects and parts using a tree-structured latent scene graph that is similar to our scene graph representation. However, GSGN learns RGB appearance models of objects and their parts, uses an inference network instead of a hybrid of data-driven and model-based inference, was not evaluated on real images or scenes, and uses more restricted scene graphs that cannot represent objects with independent 6DoF pose. GSGN builds on an earlier deep generative model [17] that generates multi-object scenes but does not model dependencies between object poses and was not quantitatively evaluated on real 3D scenes. Incorporating a learned inference network for jointly proposing scene graphs into our framework is an interesting area for future work. The term 'scene graph' has also been used in computer vision to refer to various less related graph representations of scenes [3, 8, 32].

**Probabilistic programming for computer vision** Prior work has used probabilistic programs to represent generative models of images and implemented inference in these models using probabilistic programming systems [27, 25]. Unlike these prior works, which relied on manually specified and/or semi-parametric shape models, 3DP3 learns object shapes non-parametrically. 3DP3 also models occlusion of one 3D object by another; uses a novel hierarchical scene graph prior that allows for dependencies between object poses in the prior; uses a novel involutive MCMC [10] kernel for inferring scene graph structure; and uses a novel pseudo-marginal approach for handling uncertainty about object shape during inference. We also present a proof of concept that our system can infer the presence and pose of fully occluded objects.

**6DoF object pose estimation** We use 6DoF estimation of object pose from RGBD images as an example application. Registration of point clouds [5] can be used to estimate the 6DoF pose of objects with known 3D geometry from depth images. Many recent 6DoF object pose estimators use deep learning [38, 35] and many also take depth images [37, 34]. Some pose estimation methods model scene structure, contact relationships, stability, or other semantic information [21, 8, 24, 16, 4], and some use probabilistic inference [15, 7, 19, 14]. To our knowledge, we present the first 6DoF pose estimator that uses Bayesian inference about the structure of hierarchical 3D scene graphs.

**Learning models of novel 3D objects** Classic algorithms for structure-from-motion infer a 3D model of a scene from multiple images [33, 1]. Our approach for learning the shape of novel 3D objects produces coarse-grained probabilistic voxel models of objects that can represent uncertainty about the occupancy of self-occluded volumes. Integrating other representations of object shape and object appearance [29] with our scene graph representation is a promising area of future work.

## 3   3DP3 generative modeling framework

The core of 3DP3 is a generative modeling framework that represents a scene in terms of discrete objects, the 3D shape of each object, and a hierarchical structure called a scene graph that relates the poses (position and orientation) of the objects. This section describes 3DP3's object shape and scene graph latent representations, a family of prior distributions on these latent representations, and an observation model for image-based observations of scenes. Figure 1 shows the combined generative model written as a probabilistic program.

---

[1]An early version of our work [41] is concurrent with an early version of GSGN [12]

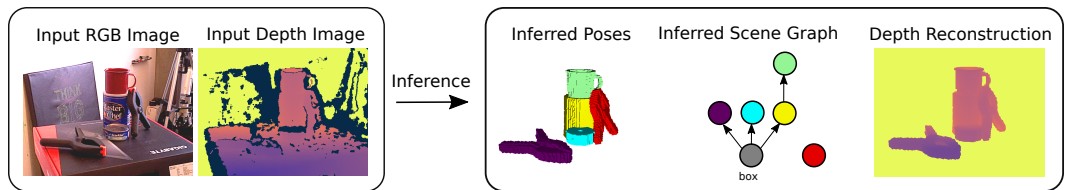

**(a)** Inferring a hierarchical 3D scene graph from an RGB-D image with 3DP3. Our model knows that objects often lay flat on other objects, which allows for the depth pixels of one object to inform the pose of other objects. Our algorithm also infers when this knowledge is relevant (e.g. the clamp on the left, represented by the purple node, is laying flat on the box), and when it is not (e.g. the clamp on the right, represented by the red node, is not laying flat on any other object).

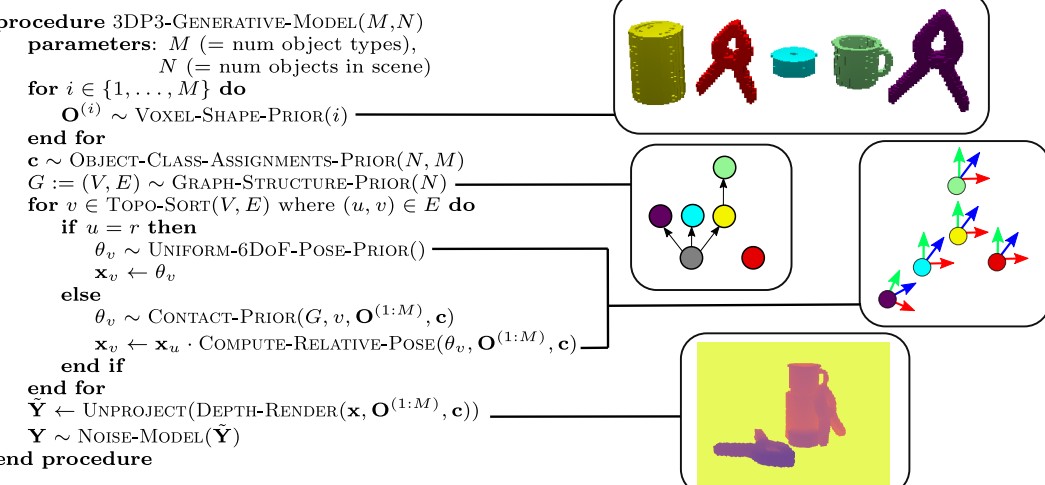

**(b)** 3DP3 uses a structured generative model of 3D scenes, represented as a probabilistic program. The model uses a prior over object shapes that can be learned from data, a prior over scene structure that is a probability distribution on graphs, a traversal of the scene graph starting at the world node $r$ to compute object poses, and a robust likelihood model for depth images. In the graph at right, the world node $r$ (not shown) is the parent of the grey node (box) and the red node (right clamp) because those objects are not layting flat on other objects.

**Figure 1:** (a) A scene graph inference task and (b) the 3DP3 generative model.

### 3.1 Objects

The most basic element of our generative modeling framework are rigid objects. The first stage in our generative model prior encodes uncertainty about the 3D shape of $M$ types of rigid objects that may or may not be encountered in any given scene.

**Voxel 3D object shapes**  We model the coarse 3D shape of rigid objects using a voxel grid with dimensions $h, w, l \in \mathbb{N}$ and cells indexed by $(i, j, \ell) \in [h] \times [w] \times [l]$. Each cell has dimension $s \times s \times s$ for resolution $s \in \mathbb{R}^+$, so that the entire voxel grid represents the cuboid $[0, h \cdot s] \times [0, w \cdot s] \times [0, l \cdot s]$. All objects are assumed to fit within the cuboid. An object's *shape* is defined by a binary assignment $\mathbf{O} \in \{0,1\}^{h \times w \times l}$ of occupancy states to each cell in the voxel grid, where $O_{ij\ell} = 1$ indicates that cell $(i, j, \ell)$ is occupied and $O_{ij\ell} = 0$ indicates it is free. Each object also has a finite set of *contact planes* through which the object may be in flush contact with the contact planes of other objects in physically stable scenes. For example, in Figure 2, the table has a contact plane for its top surface, the yellow sugar box has six contact planes, one for each of its six faces, and the bottom contact plane of the sugar box is in flush contact with the top contact plane of the table. The pose of a contact plane relative to its object is a function of the object shape $\mathbf{O}$. To simplify notation, we denote the set of contact planes for any object by $F$.

**Prior distributions on 3D object shape**  We assume there are $M$ distinct object types, and each object type $m \in \{1, \ldots, M\}$ has an a-priori unknown shape, denoted $\mathbf{O}^{(m)}$. Let $\mathbf{O}^{(1:M)} :=$

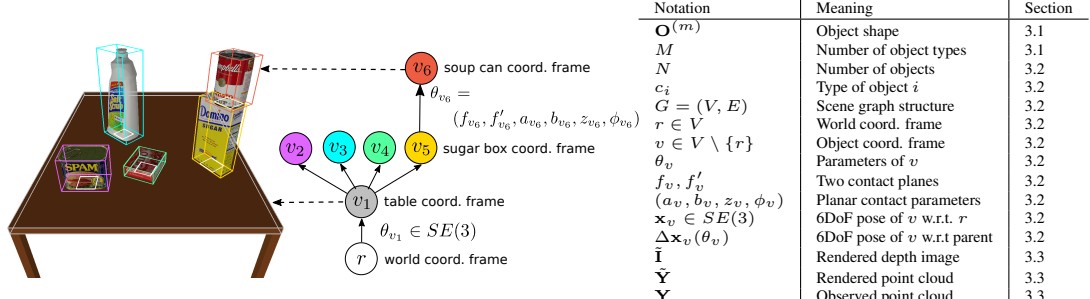

| Notation | Meaning | Section |
|---|---|---|
| $\mathbf{O}^{(m)}$ | Object shape | 3.1 |
| $M$ | Number of object types | 3.1 |
| $N$ | Number of objects | 3.2 |
| $c_i$ | Type of object $i$ | 3.2 |
| $G = (V, E)$ | Scene graph structure | 3.2 |
| $r \in V$ | World coord. frame | 3.2 |
| $v \in V \setminus \{r\}$ | Object coord. frame | 3.2 |
| $\theta_v$ | Parameters of $v$ | 3.2 |
| $f_v, f'_v$ | Two contact planes | 3.2 |
| $(a_v, b_v, z_v, \phi_v)$ | Planar contact parameters | 3.2 |
| $\mathbf{x}_v \in SE(3)$ | 6DoF pose of $v$ w.r.t. $r$ | 3.2 |
| $\Delta \mathbf{x}_v(\theta_v)$ | 6DoF pose of $v$ w.r.t parent | 3.2 |
| $\tilde{\mathbf{I}}$ | Rendered depth image | 3.3 |
| $\tilde{\mathbf{Y}}$ | Rendered point cloud | 3.3 |
| $\mathbf{Y}$ | Observed point cloud | 3.3 |

**Figure 2:** Our hierarchical scene graphs encode a tree of coordinate frames representing entities in a scene and their geometric relationships (e.g. flush contact between faces of two objects).

$(\mathbf{O}^{(1)}, \ldots, \mathbf{O}^{(M)})$. The prior distribution on the shape of each object type $m$ is denoted $p(\mathbf{O}^{(m)})$. Although our inference algorithm (Section 5) only requires the ability to sample jointly from $p(\mathbf{O}^{(1:M)})$, we assume shapes of object types are independent in the prior ($p(\mathbf{O}^{(1:M)}) = \prod_{m=1}^{M} p(\mathbf{O}^{(m)})$). Section 4 shows how to learn a specific shape prior $p(\mathbf{O}^{(m)})$ for an object type from depth images.

### 3.2 Scenes

Given a collection of $M$ known object types and their shapes, our model generates scenes with $N$ objects by randomly selecting an object type for each object and then sampling a 6DoF object pose for each object. Instead of assuming that object poses are independent, our model encodes an inductive bias about the regularities in real-world scenes: objects are often resting in flush contact with other objects (e.g. see Figure 2). We jointly sample *dependent* object poses using a flexible hierarchical scene graph, while maintaining uncertainty over the structure of the graph.

**Hierarchical scene graphs** We model the geometric state of a scene as a *scene graph* $\mathcal{G}$ (Figure 2), which is a tuple $\mathcal{G} = (G, \boldsymbol{\theta})$ where $G = (V, E)$ is a directed rooted tree and $\boldsymbol{\theta}$ are parameters. The vertices $V := \{r, v_1, \ldots, v_N\}$ represent $N + 1$ 3D coordinate frames, with $r$ representing the world coordinate frame. An edge $(u, v) \in E$ indicates that coordinate frame $v$ is parametrized relative to frame $u$, with parameters $\theta_v$. The 6DoF pose of frame $v$ relative to frame $u$ with pose $\mathbf{x}_u \in SE(3)$ is given by a function $\Delta \mathbf{x}_v$, where $\Delta \mathbf{x}_v(\theta_v) \in SE(3)$ and $\mathbf{x}_v := \mathbf{x}_u \cdot \Delta \mathbf{x}_v(\theta_v)$. Here, $\cdot$ is the $SE(3)$ group operation, and the world coordinate frame is defined as the identity element ($\mathbf{x}_r := I$).

**Modeling flush contact between rigid objects** While the vertices of scene graphs $\mathcal{G}$ can represent arbitrary coordinate frames in a scene (e.g. the coordinate frames of articulated joints, object poses), in the remainder of this paper we assume that each vertex $v \in V \setminus \{r\}$ corresponds to the pose of a rigid object. We index objects by $1, \ldots, N$, with corresponding vertices $v_1, \ldots, v_N$. We assume that each object $i$ has an object type $c_i \in \{1, \ldots, M\}$. For vertices $v$ that are children of the root vertex $r$, $\theta_v \in SE(3)$ defines the absolute 6DoF pose of the corresponding object ($\Delta \mathbf{x}_v(\theta_v) = \theta_v$). For vertices $v$ that are children of a non-root vertex $u$, the parameters take the form $\theta_v = (f_v, f'_v, a_v, b_v, z_v, \phi_v)$ and represent a contact relationship between the two objects: $f_v$ and $f'_v$ indicate which contact planes of the parent and child objects, respectively, are in contact. $(a_v, b_v) \in \mathbb{R}^2$ is the in-plane offset of the origin of plane $f_v$ of object $v$ from the origin of plane $f'_v$ of object $u$. $z_v \in \mathbb{R}$ is the perpendicular distance of the origin of plane $f_v$ of object $v$ from plane $f'_v$ of object $u$. $\phi_v \in S^2 \times S^1$ represents the deviation of the normal vectors of the two contact planes from anti-parallel (in $S^2$) and a relative in-plane rotation of the two contact planes (in $S^1$). The relative pose $\Delta \mathbf{x}_v(\theta_v)$ of $v$ with respect to $u$ is the composition (in $SE(3)$) of three relative poses: (i) $v$ with respect to its plane $f_v$, (ii) $v$'s plane $f_v$ with respect to $u$'s plane $f'_v$, and (iii) $u$'s plane $f'_v$ with respect to $u$. The 6DoF poses of all objects ($\mathbf{x}_v$ for $v \in V \setminus \{r\}$) are computed by traversing the scene graph while taking products of relative poses along paths from the root $r$.

**Prior distributions on scene graphs** We now describe our prior on scene graphs, given object models $\mathbf{O}^{(1:M)}$. We assume the number of objects $N$ in the scene is known (see the supplement for a generalization to unknown $N$). We first sample the types $c_i \in \{1, \ldots, M\}$ of all objects from an

exchangeable distribution $p(\mathbf{c})$ where $\mathbf{c} := (c_1, \ldots, c_N)$. This includes as a special case distributions where all types are represented at most once among the objects ($\sum_{i=1}^{N} \mathbf{1}[c_i = c] \leq 1$), which is the case in our experiments. Next, we sample the scene graph structure $G$ from $p(G)$. We experiment with two priors $p(G)$: (i) a uniform distribution on the set of $(N + 1)^{N-1}$ directed trees that are rooted at a vertex $r$, and (ii) $\delta_{G_0}(G)$, where $G_0$ is a graph on $N + 1$ vertices where $(r, v) \in E$ for all $v \in V \setminus \{r\}$ so that each object vertex has an independent 6DoF pose. For objects whose parent is $r$ (the world coordinate frame), we sample the pose $\theta_v \sim p_{\text{unif}}$, which samples the translation component uniformly from a cuboid scene extent, and the orientation uniformly over $SO(3)$. For objects whose parent is another object $u$, we sample the choice of contact planes $(f_v, f'_v) \in F \times F$ uniformly, $(a_v, b_v) \sim \text{Uniform}([-50\text{cm}, 50\text{cm}]^2)$, $z_v \sim \text{N}(0, 1\text{cm})$, the $S^2$ component of $\phi_v$ from a von Mises–Fisher (vMF) distribution concentrated ($\kappa = 250$) on anti-parallel plane normals, and the $S^1$ component from $\text{Uniform}(S^1)$. We denote this distribution $p_{\text{cont}}(\theta_v)$. Note that the parameters of $p_{\text{cont}}$ were not tuned or tailored in any detailed way—they were chosen heuristically based on the rough dimensions of table-top objects. The resulting prior over all of the latent variables is:

$$p(\mathbf{O}^{(1:M)}, \mathbf{c}, G, \boldsymbol{\theta}) = \left( \prod_{m=1}^{M} p(\mathbf{O}^{(m)}) \right) \frac{1}{(N+1)^{N-1}} p(\mathbf{c}) \prod_{\substack{v \in V: \\ (r,v) \in E}} p_{\text{unif}}(\theta_v) \prod_{\substack{(u,v) \in E: \\ u \neq r}} p_{\text{cont}}(\theta_v) \quad (1)$$

### 3.3 Images

Our generative model uses an observation model that generate synthetic image data given object shapes $\mathbf{O}^{(1:M)}$ and a scene graph $\mathcal{G}$ containing $N$ objects. We now describe the observation model for depth images that is used in our main experiments (Section 6).

**Likelihood model for depth images** We first convert an observed depth image into a point cloud $\mathbf{Y}$. To model a point cloud $\mathbf{Y} \in \mathbb{R}^{K \times 3}$ with $K$ points denoted $\mathbf{y}_i \in \mathbb{R}^3$, we use a likelihood model based on rendering a synthetic depth image of the scene graph. Specifically, given the object models $\mathbf{O}^{(m)}$ for each $m \in \{1, \ldots, M\}$, the object types $\mathbf{c}$, the scene graph $\mathcal{G}$, and the camera intrinsic and extrinsic parameters relative to the world frame, we (i) compute meshes from each $\mathbf{O}^{(m)}$, (ii) compute the 6DoF poses $(\mathbf{x}_v)$ of objects with respect to the world frame by traversing the scene graph $\mathcal{G}$, and (iii) render a depth image $\tilde{\mathbf{I}}$ of $\mathcal{G}$ using an OpenGL depth buffer, and (iv) unproject the rendered depth image to obtain a point cloud $\tilde{\mathbf{Y}}$ with $\tilde{K}$ points ($\tilde{K}$ is the number of pixels in the depth image). We then generate an observed point cloud $\mathbf{Y} \in \mathbb{R}^{K \times 3}$ by drawing each point from a mixture:

$$p(\mathbf{Y}|\mathbf{O}^{(1:M)}, \mathbf{c}, G, \boldsymbol{\theta}) := \prod_{i=1}^{K} \left( C \cdot \frac{1}{B} + \frac{1-C}{\tilde{K}} \sum_{j=1}^{\tilde{K}} \frac{\mathbf{1}[||\mathbf{y}_i - \tilde{\mathbf{y}}_j||_2 \leq r]}{\frac{4}{3}\pi r^3} \right) \quad (2)$$

for some $0 < C < 1$ and some $r > 0$. The components of this mixture are uniform distributions over the balls of radius $r$ centered at each point in $\tilde{\mathbf{Y}}$ (with weights $(1 - C)/\tilde{K}$) and a uniform distribution over the scene bounding volume $B$ (weight $C$).[2]

## 4 Learning object shape models

3DP3 does not require hard-coded shape models. Instead, it uses probabilistic inference to learn non-parametric models of 3D object shape $p(\mathbf{O}^{(m)})$ that account for uncertainty due to self-occlusion. We focus on the restricted setting of learning from scenes containing a single isolated object ($N = 1$) of known type ($c_1$). Our approach works best for views that lead to minimal uncertainty about the exterior shape of the object; more general, flexible treatments of shape learning and shape uncertainty are beyond the scope of this paper.

First, we group the depth images by the object type ($c_1$), so that we have $M$ independent learning problems. Let $\mathbf{I}_{1:T} := (\mathbf{I}_1, \ldots, \mathbf{I}_T)$ denote the depth observations for one object type, with object shape denoted $\mathbf{O}$. The learning algorithm uses Bayesian inference in another generative model $p'$. The posterior $p'(\mathbf{O}|\mathbf{I}_{1:T})$ produced by this algorithm becomes the prior $p(\mathbf{O})$ used in Section 3.1.

---

[2]By using a distribution that is uniform over a small, spherical region rather than a Gaussian distribution, we avoid (via k-d trees) computing pairwise distances between all points in $\mathbf{Y}$ and $\tilde{\mathbf{Y}}$, resulting in $\approx 10$x speedup.

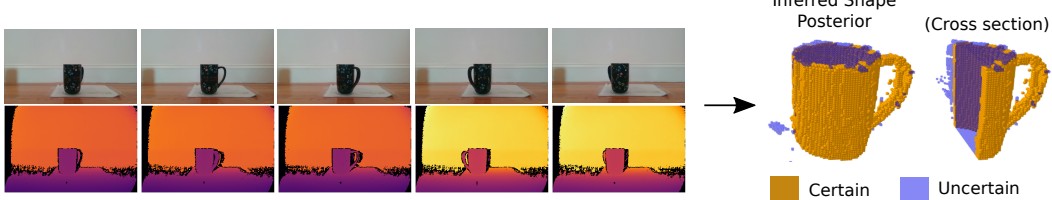

**Figure 3:** Learning a voxel-based shape models $p(\mathbf{O}^{(m)})$ for a novel object from a set of 5 depth images. Our shape priors capture uncertainty about voxel occupancy due to self-occlusion (right).

We start with a uninformed prior distribution $p'(\mathbf{O}) := \prod_{i=1}^{h} \prod_{j=1}^{w} \prod_{\ell=1}^{l} p_{\text{occ}}^{O_{ij\ell}} (1 - p_{\text{occ}})^{(1-O_{ij\ell})}$ on the 3D shape of an object type, for a per-voxel occupancy probability $p_{\text{occ}}$ (in our experiments, 0.5). We learn about the object's shape by observing a sequence of depth images $\mathbf{I}_{1:T}$ that contain views of the object, which is assumed to be static relative to other contents of the scene, which we call the 'map' $\mathbf{M}$. (In our experiments the map contains the novel object, a floor, a ceiling, and four walls of a rectangular room). We posit the following joint distribution over object shape ($\mathbf{O}$) and the observed depth images, conditioned on the map ($\mathbf{M}$) and the poses of the camera relative to the map over time $(\mathbf{x}_1, \ldots, \mathbf{x}_T \in SE(3))$: $p'(\mathbf{O}, \mathbf{I}_{1:T}|\mathbf{M}, \mathbf{x}_{1:T}) := p'(\mathbf{O}) \prod_{t=1}^{T} p'(\mathbf{I}_t|\mathbf{O}, \mathbf{M}, \mathbf{x}_t)$.

The likelihood $p'$ is a depth image likelihood on a latent 3D voxel occupancy grid (see supplement for details). For this model, we can compute $p'(\mathbf{O}|\mathbf{M}, \mathbf{x}_{1:T}, \mathbf{I}_{1:T}) = \prod_{ij\ell} p'(O_{ij\ell}|\mathbf{M}, \mathbf{x}_{1:T}, \mathbf{I}_{1:T})$ exactly using ray marching to decide if a voxel cell is occupied, unoccupied, or unobserved (due to being occluded by another occupied cell), and the resulting distribution on $\mathbf{O}$ can be compactly represented as an array of probabilities ($\in [0, 1]^{h \times w \times l}$). However, in real-world scenarios the map $\mathbf{M}$ and the camera poses $\mathbf{x}_{1:T}$ are not known with certainty. To handle this, our algorithm takes as input uncertain beliefs about $\mathbf{M}$ and $\mathbf{x}_{1:T}$ ($q_{\text{SLAM}}(\mathbf{M}, \mathbf{x}_{1:T}) \approx p'(\mathbf{M}, \mathbf{x}_{1:T}|\mathbf{I}_{1:T})$) that are produced by a separate probabilistic SLAM (simultaneous localization and mapping) module, and take the form of a weighted collection of $K$ particles $(\mathbf{M}^{(k)}, \mathbf{x}_{1:T}^{(k)})$: $q_{\text{SLAM}}(\mathbf{M}, \mathbf{x}_{1:T}) = \sum_{k=1}^{K} w_k \delta_{\mathbf{M}^{(k)}}(\mathbf{M}) \delta_{\mathbf{x}_{1:T}^{(k)}}(\mathbf{x}_{1:T})$. Various approaches to probabilistic SLAM can be used; we implemented it using sequential Monte Carlo (SMC) in Gen (more detail in supplement). From the beliefs $q_{\text{SLAM}}(\mathbf{M}, \mathbf{x}_{1:T})$ produced by SLAM, we approximate the object shape posterior via:

$$\hat{p}'(\mathbf{O}|\mathbf{I}_{1:T}) := \iint p'(\mathbf{O}|\mathbf{M}, \mathbf{x}_{1:T}, \mathbf{I}_{1:T}) q_{\text{SLAM}}(\mathbf{M}, \mathbf{x}_{1:T}) d\mathbf{M} d\mathbf{x}_{1:T} = \sum_{k=1}^{K} w_k p'(\mathbf{O}|\mathbf{M}^{(k)}, \mathbf{x}_{1:T}^{(k)}, \mathbf{I}_{1:T})$$

Note that while $p'(\mathbf{O}|\mathbf{M}^{(k)}, \mathbf{x}_{1:T}^{(k)}, \mathbf{I}_{1:T})$ for each $k$ can be compactly represented, the mixture distribution $\hat{p}'(\mathbf{O}|\mathbf{I}_{1:T})$ lacks the conditional independencies that make this possible. To produce a more compact representation of beliefs about the object's shape, we fit a variational approximation $q_\varphi(\mathbf{O})$ that assumes independence among voxels ($q_\varphi(\mathbf{O}) := \prod_{i \in [h]} \prod_{j \in [w]} \prod_{\ell \in [l]} \varphi_{ij\ell}^{O_{ij\ell}} \cdot (1 - \varphi_{ij\ell})^{(1-O_{ij\ell})}$) to $\hat{p}'(\mathbf{O}|\mathbf{I}_{1:T})$ using $\varphi^* := \arg\min_\varphi \text{KL}(\hat{p}'(\mathbf{O}|\mathbf{I}_{1:T})||q_\varphi(\mathbf{O}))$ (see supplement for details). This choice of variational family is sufficient for representing uncertainty about the occupancy of voxels in the *interior* of an object shape. Note that our shape-learning experiments did not result in significant uncertainty about the *exterior* shape of objects[3], and in the presence of such uncertainty, a less severe variational approximation may be needed for robust inference of scene graphs from depth images. Fig. 3 shows input depth images ($\mathbf{I}_{1:T}$) and resulting shape prior learned from $T = 5$ observations. After learning these shape distributions $q_{\boldsymbol{\varphi}}(\mathbf{O}) \approx \hat{p}'(\mathbf{O}|\mathbf{I}_{1:T})$ for each distinct object type, we use them as the shape priors $p(\mathbf{O}_i)$ within the generative model of Section 3. The supplement includes the results of a quantitative evaluation of the accuracy of shape learning.

---

[3]The lack of significant exterior shape uncertainty in shape-learning experiments allowed us to implement an optimization: Instead of the relative poses of an object's contact planes depending on $\mathbf{O}$ as described in Section 3, we assign each object type a set of six contact planes derived from the faces of the smallest axis-aligned bounding cuboid that completely contains all occupied voxels in one sample $\mathbf{O}$ from the learned prior $p(\mathbf{O}) := q_{\boldsymbol{\varphi}}(\mathbf{O})$.

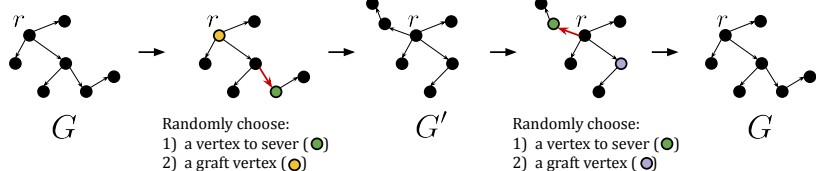

**Figure 4:** A reversible transition between scene graph structure $G$ and scene graph structure $G'$.

## 5 Building blocks for approximate inference algorithms

This section first describes a set of building blocks for approximate inference algorithms that are based on the generative model of Section 3. We then describe how to combine these components into a scene graph inference algorithm that we evaluate in Section 6.

**Trained object detectors** It is possible to infer the types of objects in the scene ($\mathbf{c}$) via Bayesian inference in the generative model (see supplement for an example that infers $\mathbf{c}$ as well as $N$ in a scene with a fully occluded object, via Bayesian inference). However, for scenes where objects are not fully or nearly-fully occluded, and where object types have dissimilar appearance, it is possible to train fast object detectors that produce an accurate point estimate of $\mathbf{c}$ given an RGB image.

**Trained pose estimators** In scenes without full or nearly-full occlusion, it is also possible to employ trained pose estimation methods [37] to give independent estimates of the 6DoF pose of each object instance in the image. However, inferring pose is more challenging than inferring $\mathbf{c}$, and occlusion, self-occlusion, and symmetries can introduce significant pose uncertainty. Therefore, we only use trained pose estimators (e.g. [37]) to (optionally) *initialize* the poses of objects before Bayesian inference in the generative model, using the building blocks below.

**Data-driven Metropolis-Hastings kernels on object pose** We employ Metropolis-Hastings (MH) kernels, parametrized by choice of object $i \in \{1, \ldots, N\}$, that take as input a scene graph $\mathcal{G}$, propose new values ($\theta'_{v_i}$) for the scene graph parameters of object $i$, construct a new proposed scene graph $\mathcal{G}'$, and then accept or reject the move from $\mathcal{G}$ to $\mathcal{G}'$ based on the MH rule. For objects $v$ whose parent is the world frame ($(r, v) \in E$), we use a data-driven proposal distribution centered on an estimate ($\hat{\mathbf{x}}_v$) of the 6DoF object pose obtained with ICP (a spherical normal distribution concentrated around the estimated position, and a vMF distribution concentrated around the estimated orientation). We also use kernels with random-walk proposals centered on the current pose. For objects whose parent is another object ($(u, v) \in E$ for $u \neq r$), we use a random-walk proposal on parameters $(a_{v_i}, b_{v_i}, z_{v_i})$. Note that when the pose of an object is changed in the proposed graph $\mathcal{G}'$, the pose of any descendant objects is also changed.[4] Each of these MH kernels is invariant with respect to $p(G, \boldsymbol{\theta}|\mathbf{c}, \mathbf{Y})$.

**Involutive MCMC kernel on scene graph structure** To infer the scene graph structure $G$, we employ a family of involutive MCMC kernels [10] that propose a new graph structure $G'$ while keeping the poses ($\mathbf{x}_v$) of all objects fixed. The kernel takes a graph structure $G$ and proposes a new graph structure $G'$ (Figure 4) by: (i) randomly sampling a node $v \in V \setminus \{r\}$ to 'sever' from the tree, (ii) randomly choosing a node $u \in V \setminus \{v\}$ that is not a descendant of the severed node on which to graft $v$, (iii) forming a new directed graph $G'$ over vertices $V$ by grafting $v$ to $u$; by Lemma O.7.1 the resulting graph $G'$ is also a tree. Note that there is an involution $g$ on the set of all pairs $(G, v, u)$ satisfying the above constraints. That is, if $(G', v', u') = g(G, v, u)$ then $(G, v, u) = g(G', v', u')$. (This implies, for example, that $u'$ is the parent of $v$ in $G$.) Note that this set of transitions is capable of changing the parent vertex of an object to a different parent object, changing the parent vertex of an object from the root (world frame) to any other object, or changing the parent vertex from another object to the root, depending on the random choice of $v$ and $u$. We compute new values for parameters ($\theta_v$) for the severed node $v$ and possibly other vertices such that the poses of all vertices are unchanged. See supplement for the full kernel and a proof that it is invariant w.r.t. $p(G, \boldsymbol{\theta}|\mathbf{c}, \mathbf{Y})$.

---

[4]It is possible to construct an involutive MCMC kernel that does not change the poses of descendant objects.

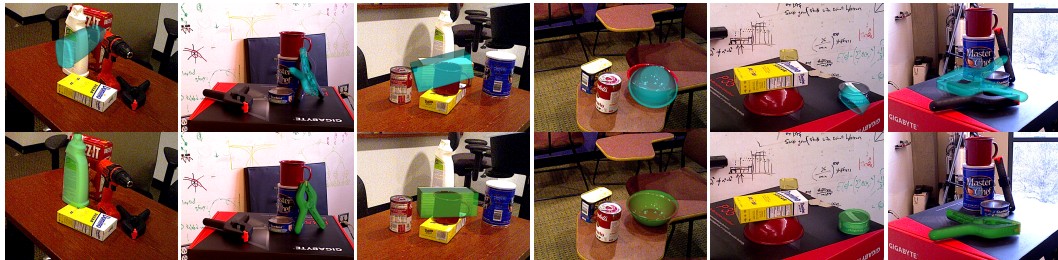

**Figure 5:** Qualitative comparison between DenseFusion's pose estimates (top row) and estimates from 3DP3-based algorithm that is initialized with DenseFusion (bottom row) for YCB-Video frames where DenseFusion gives incorrect results. 3DP3's depth-rendering likelihood and scene graph prior can correct large errors made by DenseFusion.

**Approximately Rao–Blackwellizing object shape via pseudo-marginal MCMC** The acceptance probability expressions for our involutive MCMC and MH kernels targeting $p(G, \boldsymbol{\theta} | \mathbf{c}, \mathbf{Y})$ include factors of the form $p(\mathbf{Y} | \mathbf{c}, G, \boldsymbol{\theta})$, which is an intractable sum over the latent object models: $p(\mathbf{Y} | \mathbf{c}, G, \boldsymbol{\theta}) = \sum_{\mathbf{O}^{(1:M)}} p(\mathbf{O}^{(1:M)}) p(\mathbf{Y} | \mathbf{O}^{(1:M)}, \mathbf{c}, G, \boldsymbol{\theta})$. To overcome this challenge, we employ a pseudo-marginal MCMC approach [2] that uses unbiased estimates of $p(\mathbf{Y} | \mathbf{c}, G, \boldsymbol{\theta})$ obtained via likelihood weighting (that is, sampling several times from $p(\mathbf{O}^{(1:M)})$ and averaging the resulting $p(\mathbf{Y} | \mathbf{O}^{(1:M)}, \mathbf{c}, G, \boldsymbol{\theta})$). The resulting MCMC kernels are invariant with respect to an extended target distribution of which $p(G, \boldsymbol{\theta} | \mathbf{c}, \mathbf{Y})$ is a marginal (see supplement for details). We implemented an optimization where we sampled 5 values for $\mathbf{O}^{(1:M)}$ and used these samples within every estimate of $p(\mathbf{Y} | \mathbf{c}, G, \boldsymbol{\theta})$ instead of sampling new values for each estimate. Because our learned shape priors did not have significant exterior shape uncertainty, this optimization did not negatively impact the results.

**Scene graph inference and implementation** The end-to-end scene graph inference algorithm has three stages. First, we obtain $\mathbf{c}$ from either an object detector or because it is given as part of the task (this is the case in our experiments; see Section 6 for details). Second, we obtain initial estimates $\hat{\mathbf{x}}_v$ of 6DoF object poses $\mathbf{x}_v$ for all object vertices $v$ via maximum-a-posteriori (MAP) inference in a restricted variant of the generative model with graph structure $G$ fixed to $G_0$ (so there are no edges between object vertices). This MAP inference stage uses the data-driven Metropolis-Hastings kernels on poses, and (optionally) trained pose estimators (see Section 6 for the details, which differ between experiments). Third, we use the estimated poses to initialize an MCMC algorithm targeting $p(G, \boldsymbol{\theta} | \mathbf{c}, \mathbf{Y})$ with state $G \leftarrow G_0$ and $\theta_v \leftarrow \hat{\mathbf{x}}_v$ for each $v \in V \setminus \{r\}$. The Markov chain is a cycle of the involutive MCMC kernel described above with a mixture of the Metropolis-Hastings kernels described above, uniformly mixed over objects. We wrote the probabilistic program of Figure 1 in Gen's built-in modeling language. We implemented the data-driven and involutive MCMC kernels, and pseudo-marginal likelihood, and integrated all components together, using Gen's programmable inference support. Our code is available at `https://github.com/probcomp/ThreeDP3`.

## 6 Experiments

We evaluate our scene graph inference algorithm on the YCB-Video [6] dataset consisting of real RGB-D images and YCB-Challenging, our own synthetic dataset of scenes containing novel viewpoints, occlusions, and contact structure. We use the evaluation protocol of the Benchmark for 6DoF Object Pose Estimation (BOP) Challenge [20], in which an RGB-D image and the number of objects in the scene and their types are given, and the task is to estimate the 6DoF pose of each object.

### 6.1 Pose estimation from real RGB-D images

YCB-Video is a standard robotics dataset for training and evaluating 3D perception systems [6]. We first learn shape priors (Section 4) from just 5 synthetic images for each object type. We use DenseFusion [37], a neural 6DoF pose estimator, for pose initialization in the MAP phase of our inference algorithm. To measure pose estimation accuracy, we use the average closest point distance (ADD-S [38, 37]) which estimates the average closest point distance between points on the object model placed at the predicted pose and points on the model placed at the ground-truth pose. Table 1

| Object Type | # of Scenes | 0.5cm Threshold | | | 1.0cm Threshold | | | 2.0cm Threshold | | |
|---|---|---|---|---|---|---|---|---|---|---|
| | | **3DP3** | **Accuracy 3DP3\*** | **DF** | **3DP3** | **Accuracy 3DP3\*** | **DF** | **3DP3** | **Accuracy 3DP3\*** | **DF** |
| 002_master_chef_can | 1006 | 0.74 | 0.79 | **0.84** | 0.99 | 1.00 | **1.00** | **1.00** | **1.00** | **1.00** |
| 003_cracker_box | 868 | **0.90** | 0.83 | 0.79 | **0.99** | 0.98 | 0.97 | 0.99 | 0.99 | **0.99** |
| 004_sugar_box | 1182 | **1.00** | 0.99 | 0.98 | **1.00** | **1.00** | **1.00** | **1.00** | **1.00** | **1.00** |
| 005_tomato_soup_can | 1440 | **0.95** | 0.93 | 0.93 | **0.97** | 0.97 | 0.97 | **0.97** | **0.97** | **0.97** |
| 006_mustard_bottle | 357 | **0.99** | 0.98 | 0.94 | **0.99** | **0.99** | 0.98 | **1.00** | **1.00** | **1.00** |
| 007_tuna_fish_can | 1148 | 0.81 | 0.80 | **0.91** | **1.00** | **1.00** | 0.99 | **1.00** | **1.00** | **1.00** |
| 008_pudding_box | 214 | **1.00** | 0.97 | 0.70 | **1.00** | 1.00 | 1.00 | **1.00** | **1.00** | **1.00** |
| 009_gelatin_box | 214 | **1.00** | **1.00** | **1.00** | **1.00** | **1.00** | **1.00** | **1.00** | **1.00** | **1.00** |
| 010_potted_meat_can | 766 | **0.80** | 0.78 | 0.79 | **0.89** | 0.88 | 0.87 | **0.93** | 0.93 | 0.92 |
| 011_banana | 379 | **0.98** | 0.96 | 0.82 | **1.00** | **1.00** | 0.97 | **1.00** | **1.00** | 1.00 |
| 019_pitcher_base | 570 | **1.00** | 0.99 | 0.99 | **1.00** | 1.00 | 1.00 | **1.00** | **1.00** | **1.00** |
| 021_bleach_cleanser | 1029 | **0.94** | 0.88 | 0.80 | **1.00** | **1.00** | 0.99 | **1.00** | **1.00** | 1.00 |
| 024_bowl | 406 | **0.93** | 0.87 | 0.50 | **0.96** | **0.96** | 0.56 | **0.96** | 0.96 | 0.94 |
| 025_mug | 636 | **0.89** | 0.89 | 0.92 | 0.98 | 0.98 | **0.99** | **1.00** | **1.00** | **1.00** |
| 035_power_drill | 1057 | **0.98** | 0.96 | 0.88 | **0.99** | 0.99 | 0.98 | **0.99** | 0.99 | 0.99 |
| 036_wood_block | 242 | **0.36** | 0.33 | 0.07 | **0.96** | 0.93 | 0.88 | **1.00** | **1.00** | **1.00** |
| 037_scissors | 181 | **0.75** | 0.69 | 0.20 | **0.87** | 0.84 | 0.70 | 0.99 | **0.99** | 0.98 |
| 040_large_marker | 648 | **1.00** | **1.00** | 0.99 | **1.00** | **1.00** | **1.00** | **1.00** | **1.00** | **1.00** |
| 051_large_clamp | 712 | **0.68** | 0.64 | 0.25 | **0.71** | 0.70 | 0.33 | **0.79** | 0.79 | 0.79 |
| 052_extra_large_clamp | 682 | **0.33** | 0.27 | 0.12 | **0.38** | 0.34 | 0.17 | 0.69 | 0.70 | **0.74** |
| 061_foam_brick | 288 | **0.26** | 0.24 | 0.01 | **1.00** | **1.00** | 0.99 | **1.00** | **1.00** | **1.00** |

**Table 1:** Accuracy results on the real YCB-Video test set, for accuracy thresholds 0.5cm, 1.0cm, and 2.0cm, and per object type. 3DP3 is our full scene graph inference algorithm and 3DP3* is an ablation that does not infer contact relationships. '# of Scenes' = The number of test images in which that object appears, out of the total 2,949 images. 'DF' = DenseFusion [37], a deep learning baseline.

shows the quantitative results. For almost all objects, our algorithm (3DP3) is more accurate than an ablation (3DP3*) that fixes the structure so that there are no contact relationships, and the ablation is more accurate than DenseFusion. This suggests that both the rendering-based likelihood and inference of structure contribute to 3DP3's more accurate 6DoF pose estimation. Figure 5 shows examples of corrections that 3DP3 makes to DenseFusion's estimates.

## 6.2 Generalization to challenging scenes

Next, we evaluated our algorithm's performance on challenging scenes containing novel viewpoints, occlusions, and contact structure. Our synthetic YCB-Challenging dataset consists of 2000 RGB-D images containing objects from the YCB object set [6] in the following 4 categories of challenging scenes: (i) *Single object*: Single object in contact with table, (ii) *Stacked*: Stack of two objects on a table, (iii) *Partial view*: Single object not fully in field-of-view, (iv) *Partially Occluded*: One object partially occluded by another. For this experiment, the MAP stage of our algorithm uses an alternative initialization (see supplement) that does not use DenseFusion. We evaluate 3DP3 and the 3DP3* ablation alongside DenseFusion [37] and another state-of-the-art baseline, Robust6D [34]. For most scenes and objects, our approach significantly outperforms the baselines (Table 2). In Table 3, we assess 3DP3's robustness by inspecting the error distribution at the 1st, 2nd, and 3rd quartile for each scene type and object type. At Q3, 3DP3 consistently outperforms the baselines and we find that the drop in performance from Q1 and Q3 is less for 3DP3 than the baselines.

## 7 Discussion

This paper presented 3DP3, a framework for generative modeling, learning, and inference with structured scenes and image data; and showed that it improves the accuracy of 6DoF object pose estimation in cluttered scenes. We used probabilistic programs to conceive of our generative model and represent it concisely; and we used a probabilistic programming system [11] with programmable inference [28] to manage the complexity of our inference and learning algorithm implementations. The current work has several limitations: Our algorithm runs $\approx$ 20x slower than the DenseFusion baseline. Our shape-learning algorithm requires that the training scenes contain only the single novel object, whose identity is known across training frames. Adding the ability to segment and learn models of novel objects in cluttered scenes and automatically train object detectors and pose estimators for these objects from short RGB-D video sequences, is an ongoing direction of work. The model also does not yet incorporate some important prior knowledge about scenes—interpenetration of objects is permitted, and constraints on physical stability are not incorporated. More experiments are also needed to understand the implications of a Bayesian treatment of 3D scene perception.

| Scene Type | Object Type | # of Scenes | 0.5cm Threshold | | | | 1.0cm Threshold | | | | 2.0cm Threshold | | | |
|---|---|---|---|---|---|---|---|---|---|---|---|---|---|---|
| | | | 3DP3 | Accuracy 3DP3* | DF | R6D | 3DP3 | Accuracy 3DP3* | DF | R6D | 3DP3 | Accuracy 3DP3* | DF | R6D |
| **Single Object** | 002_master_chef_can | 94 | **0.99** | 0.95 | 0.45 | 0.03 | **1.00** | **1.00** | 0.69 | 0.46 | **1.00** | **1.00** | **1.00** | 0.98 |
| | 003_cracker_box | 92 | **0.55** | 0.39 | 0.16 | 0.00 | **0.98** | **0.98** | 0.39 | 0.02 | **1.00** | **1.00** | 0.78 | 0.42 |
| | 004_sugar_box | 109 | **0.90** | 0.87 | 0.17 | 0.00 | **1.00** | **1.00** | 0.72 | 0.32 | **1.00** | **1.00** | **1.00** | **1.00** |
| | 005_tomato_soup_can | 108 | **0.88** | 0.81 | 0.18 | 0.00 | **1.00** | **1.00** | 0.36 | 0.07 | **1.00** | **1.00** | 0.86 | 0.74 |
| | 006_mustard_bottle | 97 | **0.86** | 0.79 | 0.48 | 0.01 | **1.00** | **1.00** | 0.57 | 0.36 | **1.00** | **1.00** | 0.81 | 0.89 |
| **Stacked** | 002_master_chef_can | 190 | **0.86** | 0.79 | 0.28 | 0.02 | **0.94** | 0.93 | 0.56 | 0.39 | 0.95 | 0.95 | **1.00** | 0.98 |
| | 003_cracker_box | 204 | **0.41** | 0.24 | 0.16 | 0.00 | **0.85** | 0.81 | 0.41 | 0.04 | **0.97** | 0.96 | 0.76 | 0.40 |
| | 004_sugar_box | 214 | **0.63** | 0.61 | 0.14 | 0.01 | **0.92** | 0.91 | 0.61 | 0.33 | 0.94 | 0.94 | 0.99 | **0.99** |
| | 005_tomato_soup_can | 193 | **0.67** | 0.52 | 0.13 | 0.00 | **0.89** | 0.86 | 0.28 | 0.06 | **0.90** | 0.88 | 0.75 | 0.66 |
| | 006_mustard_bottle | 199 | **0.73** | 0.60 | 0.44 | 0.03 | **0.94** | 0.90 | 0.54 | 0.30 | **0.94** | **0.94** | 0.85 | 0.88 |
| **Partial View** | 002_master_chef_can | 106 | **0.81** | 0.80 | 0.11 | 0.00 | **1.00** | **1.00** | 0.30 | 0.04 | **1.00** | **1.00** | 0.67 | 0.42 |
| | 003_cracker_box | 99 | **0.18** | 0.16 | 0.00 | 0.00 | **0.60** | 0.57 | 0.01 | 0.00 | **0.82** | 0.80 | 0.14 | 0.04 |
| | 004_sugar_box | 111 | **0.63** | 0.59 | 0.00 | 0.00 | **0.89** | 0.89 | 0.08 | 0.04 | **1.00** | **1.00** | 0.73 | 0.68 |
| | 005_tomato_soup_can | 87 | **0.34** | 0.33 | 0.00 | 0.00 | **0.72** | 0.71 | 0.13 | 0.00 | **0.83** | 0.82 | 0.40 | 0.13 |
| | 006_mustard_bottle | 97 | 0.55 | **0.62** | 0.08 | 0.00 | **0.87** | 0.86 | 0.23 | 0.00 | **0.96** | 0.95 | 0.37 | 0.26 |
| **Partially Occluded** | 002_master_chef_can | 130 | **0.71** | 0.52 | 0.04 | 0.00 | **0.93** | 0.90 | 0.13 | 0.02 | **0.99** | **0.99** | 0.22 | 0.12 |
| | 003_cracker_box | 500 | 0.37 | 0.35 | **0.59** | 0.00 | **1.00** | **1.00** | **1.00** | 0.02 | **1.00** | **1.00** | **1.00** | 1.00 |
| | 004_sugar_box | 117 | 0.02 | 0.01 | **0.06** | 0.00 | 0.30 | 0.27 | **0.40** | 0.12 | **0.94** | 0.93 | 0.84 | 0.77 |
| | 005_tomato_soup_can | 124 | **0.04** | 0.00 | 0.01 | 0.00 | **0.31** | 0.23 | 0.14 | 0.06 | **0.81** | 0.75 | 0.50 | 0.49 |
| | 006_mustard_bottle | 129 | **0.70** | 0.43 | 0.55 | 0.03 | 0.84 | 0.74 | **0.95** | 0.29 | 0.94 | 0.90 | **1.00** | 0.99 |

**Table 2:** Accuracy results on our synthetic YCB-Challenging data set. We report the number of scenes over which this accuracy is computed for each object and scene type. Accuracy is shown for 3DP3 and 3DP3*, which are our full method and an ablation that does not model contact relationships, respectively, and two deep learning baselines (DenseFusion (DF) [37] and Robust6D (R6D) [34]).

| Scene Type | Method | Tomato Soup | | | Cracker Box | | | Potted Meat | | | Sugar Box | | | Master Chef | | |
|---|---|---|---|---|---|---|---|---|---|---|---|---|---|---|---|---|
| | | ADD-S Q1 | Q2 | Q3 | ADD-S Q1 | Q2 | Q3 | ADD-S Q1 | Q2 | Q3 | ADD-S Q1 | Q2 | Q3 | ADD-S Q1 | Q2 | Q3 |
| **Single object** | 3DP3 (ours) | **0.35** | **0.39** | **0.41** | **0.43** | **0.49** | **0.54** | **0.36** | 0.40 | 0.45 | **0.38** | **0.43** | **0.48** | 0.37 | 0.43 | **0.48** |
| | 3DP3* (ours) | 0.35 | 0.40 | 0.43 | 0.47 | 0.52 | 0.62 | 0.36 | **0.39** | **0.44** | 0.40 | 0.45 | 0.49 | 0.35 | **0.41** | 0.49 |
| | DenseFusion | 0.35 | 0.55 | 1.11 | 0.65 | 1.35 | 1.72 | 0.58 | 0.88 | 1.02 | 0.67 | 1.25 | 1.72 | **0.32** | 0.61 | 1.85 |
| | Robust6D | 0.84 | 1.05 | 1.29 | 1.65 | 2.22 | 2.90 | 0.97 | 1.09 | 1.21 | 1.25 | 1.61 | 2.02 | 0.83 | 1.48 | 1.89 |
| **Stacked** | 3DP3 (ours) | **0.37** | **0.42** | **0.46** | **0.46** | **0.52** | **0.60** | **0.39** | **0.45** | **0.60** | **0.40** | **0.46** | **0.52** | 0.40 | **0.45** | **0.51** |
| | 3DP3* (ours) | 0.38 | 0.42 | 0.48 | 0.50 | 0.60 | 0.79 | 0.40 | 0.46 | 0.64 | 0.43 | 0.49 | 0.61 | 0.41 | 0.47 | 0.56 |
| | DenseFusion | 0.49 | 0.87 | 1.21 | 0.66 | 1.33 | 1.97 | 0.63 | 0.92 | 1.16 | 0.93 | 1.42 | 1.96 | **0.37** | 0.66 | 1.83 |
| | Robust6D | 0.84 | 1.15 | 1.36 | 1.68 | 2.21 | 2.86 | 0.92 | 1.11 | 1.26 | 1.37 | 1.72 | 2.21 | 0.94 | 1.38 | 1.82 |
| **Partial view** | 3DP3 (ours) | 0.34 | **0.40** | **0.47** | **0.54** | **0.76** | 1.56 | **0.36** | **0.45** | **0.59** | 0.47 | **0.55** | 1.80 | 0.36 | 0.47 | **0.57** |
| | 3DP3* (ours) | **0.33** | 0.40 | 0.47 | 0.56 | 0.90 | **1.54** | 0.37 | 0.45 | 0.63 | **0.46** | 0.59 | 1.81 | 0.36 | **0.46** | 0.58 |
| | DenseFusion | 0.79 | 1.52 | 2.10 | 2.33 | 2.93 | 3.81 | 1.26 | 1.65 | 2.07 | 1.52 | 2.14 | 2.78 | 1.05 | 2.22 | 2.71 |
| | Robust6D | 1.43 | 2.25 | 2.93 | 3.40 | 4.03 | 4.77 | 1.51 | 1.83 | 2.13 | 2.24 | 2.99 | 4.30 | 1.97 | 2.50 | 3.27 |
| **Partially Occluded** | 3DP3 (ours) | **0.36** | **0.42** | 0.52 | 0.48 | 0.52 | **0.55** | 0.91 | 1.25 | 1.57 | **0.89** | **1.69** | **1.97** | 0.36 | **0.42** | **0.55** |
| | 3DP3* (ours) | 0.39 | 0.49 | 0.64 | 0.48 | 0.53 | 0.58 | 0.97 | 1.29 | 1.66 | 1.03 | 1.72 | 1.99 | 0.43 | 0.58 | 1.01 |
| | DenseFusion | – | – | – | **0.41** | **0.48** | 0.58 | **0.81** | **1.11** | **1.53** | 1.47 | 2.01 | 3.18 | 0.38 | 0.48 | 0.64 |
| | Robust6D | – | – | – | 1.30 | 1.48 | 1.61 | 1.15 | 1.46 | 1.87 | 1.48 | 2.02 | 3.24 | 0.94 | 1.10 | 1.33 |

**Table 3:** Robustness of inference. We quantify the ADD-S error at 1st, 2nd, and 3rd quartiles for each scene type and object type in the synthetic dataset of hard scenes. A value of – indicates the method made no prediction for the object's pose. 3DP3* denotes an ablated version of our method without inference of the scene graph structure and thus object-object contact.

# 8 Acknowledgements

The authors acknowledge Javier Felip Leon (Intel) for helpful discussions and a prototype depth renderer, and Omesh Tickoo (Intel) for helpful discussions. This work was funded in part by the DARPA Machine Common Sense program (Award ID: 030523-00001); by the Singapore DSTA / MIT SCC collaboration; by Intel's Probabilistic Computing Center; and by philanthropic gifts from the Aphorism Foundation and the Siegel Family Foundation. We thank Alex Lew, Tan Zhi-Xuan, Feras Saad, Cameron Freer, McCoy Becker, Sam Witty, and George Matheos for helpful feedback.

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
