# Supplemental Material for
# 3DP3: 3D Scene Perception via Probabilistic Programming

**Nishad Gothoskar**[1]         **Marco Cusumano-Towner**[1]         **Ben Zinberg**[1]

**Matin Ghavamizadeh**[1]         **Falk Pollok**[2]         **Austin Garrett**[1]

**Joshua B. Tenenbaum**[1]         **Dan Gutfreund**[2]         **Vikash K. Mansinghka**[1]

[1]MIT         [2]MIT-IBM Watson AI Lab
{nishad,marcoct,bzinberg,mghavami,jbt,vkm}@mit.edu
{falk.pollok,austin.garrett}@ibm.com     dgutfre@us.ibm.com

## A   Broader Impact

While the goal of robust scene parsing and pose estimation is challenging, and the present work is an early step with much more work lying ahead, it is important to consider potential societal impacts of this work, both positive and negative. Robust pose estimation will be instrumental in improving the reliability of a wide variety of applications—including assistive technologies for people with limited mobility, improved fault detection in manufacturing plants, and safer autopilot for autonomous vehicles. On the other hand, these same technologies, if used toward the wrong ends, could have negative societal impacts as well, such as unjust or inequitable surveillance, or weapon guidance systems that fall into the wrong hands. Even applications that are largely beneficial must be implemented thoughtfully to avoid negative side effects. For example, in the present work, the choice of prior distribution on contact structures implies an inductive bias that, if chosen incorrectly, could lead to technologies that are less reliable when the scene being parsed contains a person in a wheelchair. As a scientific community, it is important that we place continued emphasis on developing technical safeguards against both overt misuse and unintended consequences like the above. Furthermore, we must remember that technical safeguards on their own are not sufficient: we must communicate to broader society not just the benefits, but also the risks of this technology, so that users can be informed participants and apply this technology towards a better world.

## B   Pose estimation from synthetic RGB images

In the previous two sections, 3DP3 was used with a depth-rendering-based likelihood on depth images since an RGB-D image was given as input. In this section, we show that 3DP3 can be used to do pose estimation without depth data i.e. from just an RGB image. Instead of a depth likelihood, we substitute an RGB renderer and simple color likelihood. We qualitatively compare with Attend, Infer, Repeat (AIR) [7], an amortized inference approach based on recurrent neural networks which can be applied to infer poses of 3D objects. We generated scenes that resemble the tabletop scenes on which AIR qualitatively assessed pose inference accuracy. Figure 1 shows pairs of input RGB images and corresponding reconstructions from pose inferences made by 3DP3. Qualitatively, our system produces pose inferences of better or equal accuracy to AIR. Importantly, our system does not require training. In contrast, AIR takes approximately 3 days for training to converge. Also at these lower resolutions, our inference can run in 0.5s per frame.

35th Conference on Neural Information Processing Systems (NeurIPS 2021).

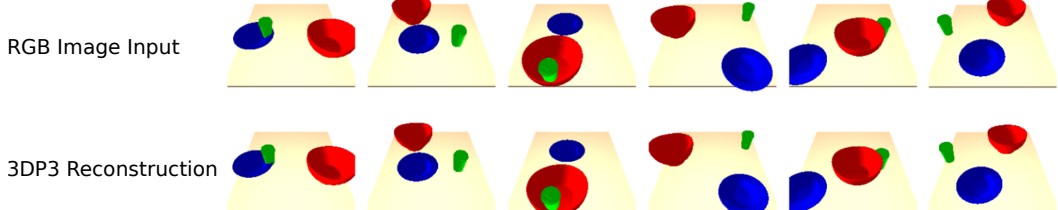

RGB Image Input

3DP3 Reconstruction

**Figure 1:** A variant of our scene graph inference algorithm that uses a RGB-based likelihood applied to synthetic RGB scenes designed to resemble those used in the evaluation of AIR [7]. Our algorithm gives accurate reconstructions with 0.5 seconds of inference time on these scenes and no training.

## C Shape Learning Accuracy Quantitative Evaluation

| Object Type | IoU |
|---|---|
| 002_master_chef_can | 0.9544 |
| 003_cracker_box | 0.9716 |
| 004_sugar_box | 0.9484 |
| 005_tomato_soup_can | 0.9433 |
| 006_mustard_bottle | 0.9671 |
| 007_tuna_fish_can | 0.9696 |
| 008_pudding_box | 0.9617 |
| 009_gelatin_box | 0.9451 |
| 010_potted_meat_can | 0.9654 |
| 011_banana | 0.9599 |
| 019_pitcher_base | 0.9808 |
| 021_bleach_cleanser | 0.9582 |
| 024_bowl | 0.9694 |
| 025_mug | 0.9621 |
| 035_power_drill | 0.966 |
| 036_wood_block | 0.9679 |
| 037_scissors | 0.9505 |
| 040_large_marker | 0.9767 |
| 051_large_clamp | 0.9218 |
| 052_extra_large_clamp | 0.9228 |
| 061_foam_brick | 0.9405 |

**Table 1:** We include a quantitative evaluation comparing the learned shape models to the ground truth shape models. To get a shape model from the learned shape prior, we take all voxels which the prior says are more likely to be occupied than unoccupied and compute the IoU between that volume and the ground truth object volume.

## D YCB-Challenging Dataset

YCB-Challenging is a synthetic test dataset of 2000 RGB-D images, 500 in each of the following 4 categories:

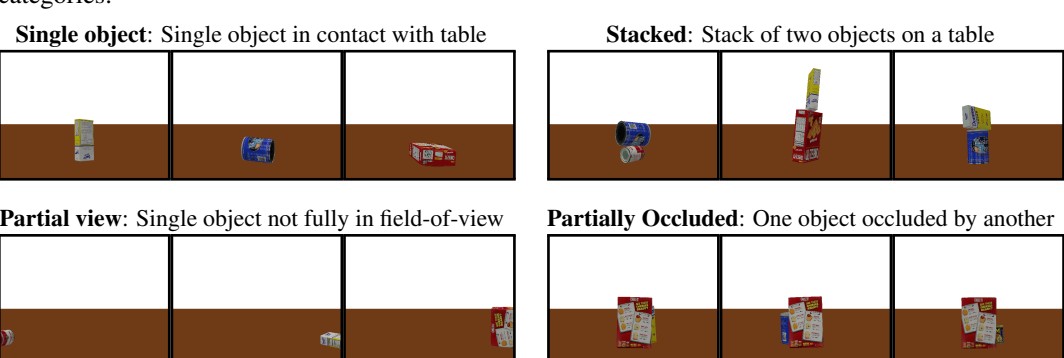

**Single object**: Single object in contact with table

**Stacked**: Stack of two objects on a table

**Partial view**: Single object not fully in field-of-view

**Partially Occluded**: One object occluded by another

# E    YCB-Challenging Extended Experimental Results

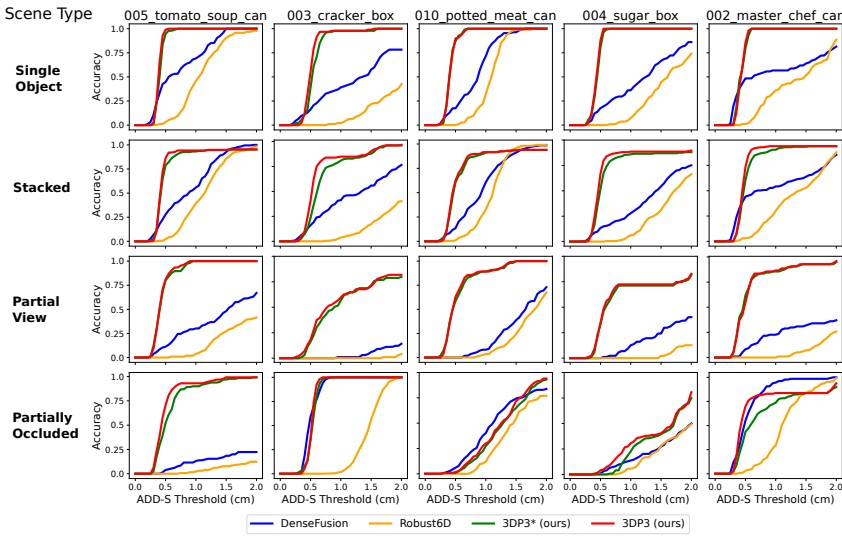

**Figure 2:** Accuracy of our method and two deep learning baselines (DenseFusion [13] and Robust6D [12]) on the task of 6DoF pose estimation in our synthetic 'YCB-Challenging' dataset. For each of the 4 scene types (rows) and 5 object types (columns), we measure accuracy for a range of ADD-S thresholds. 3DP3* denotes an ablated version of our method without inference of the scene graph structure and thus object-object contact (i.e. we fix the scene graph to $G_0$)

# F    YCB-Video Dataset

## YCB-Video Test Data Set

| Scene 1 | Scene 2 | ... | Scene 12 |
|---|---|---|---|
| 025_mug | 024_bowl | | 003_cracker_box |
| 002_master_chef_can | 004_sugar_box | | 005_tomato_soup_can |
| 051_large_clamp | 007_tuna_fish_can | | 007_tuna_fish_can |
| 007_tuna_fish_can | 010_potted_meat_can | | 010_potted_meat_can |
| 052_extra_large_clamp | | | 035_power_drill |
| | | | 040_large_marker |

2,949 total images

**Figure 3:** The YCB-Video [3] test data set consists of 2,949 real RGB-D images featuring the 21 YCB objects. These 2,949 images are collected from videos of 12 different scenes where the camera pans around the scene to view it from different perspectives. The 12 scenes contain different subsets of the 21 YCB objects, and some objects appear in multiple scenes (e.g 007_tuna_fish_can appears in Scenes 1, 2, and 12).

# G    Ablation Qualitative Results on YCB-Video

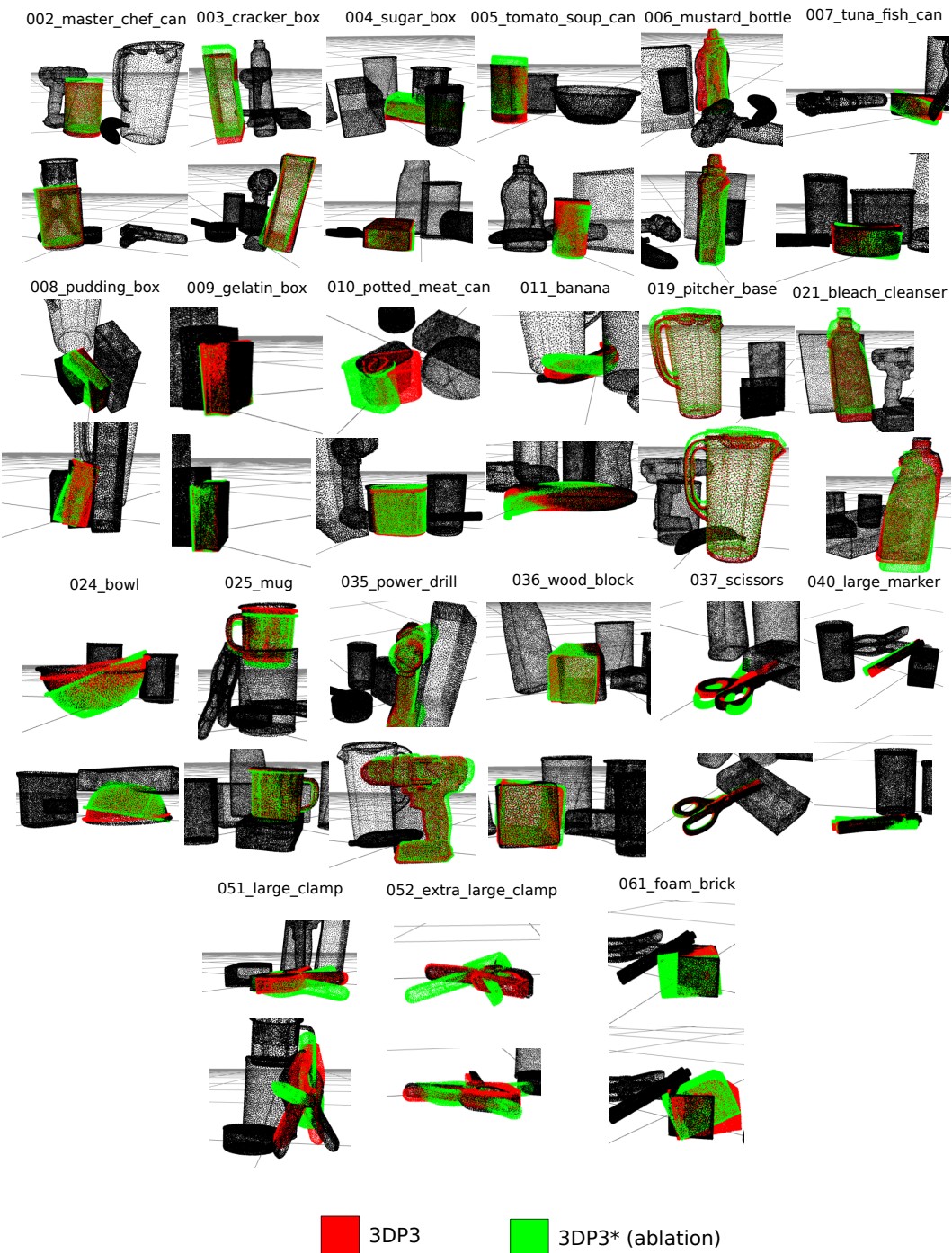

**Figure 4:** Comparison with ablated model on YCB-V real scenes. For each of the 21 YCB objects, we show 2 images of scenes containing that object and the poses estimated by our full method (3DP3) and an ablated version of our method (3DP3*) that does not model contact relationships.

# H    Qualitative Results on YCB-Video

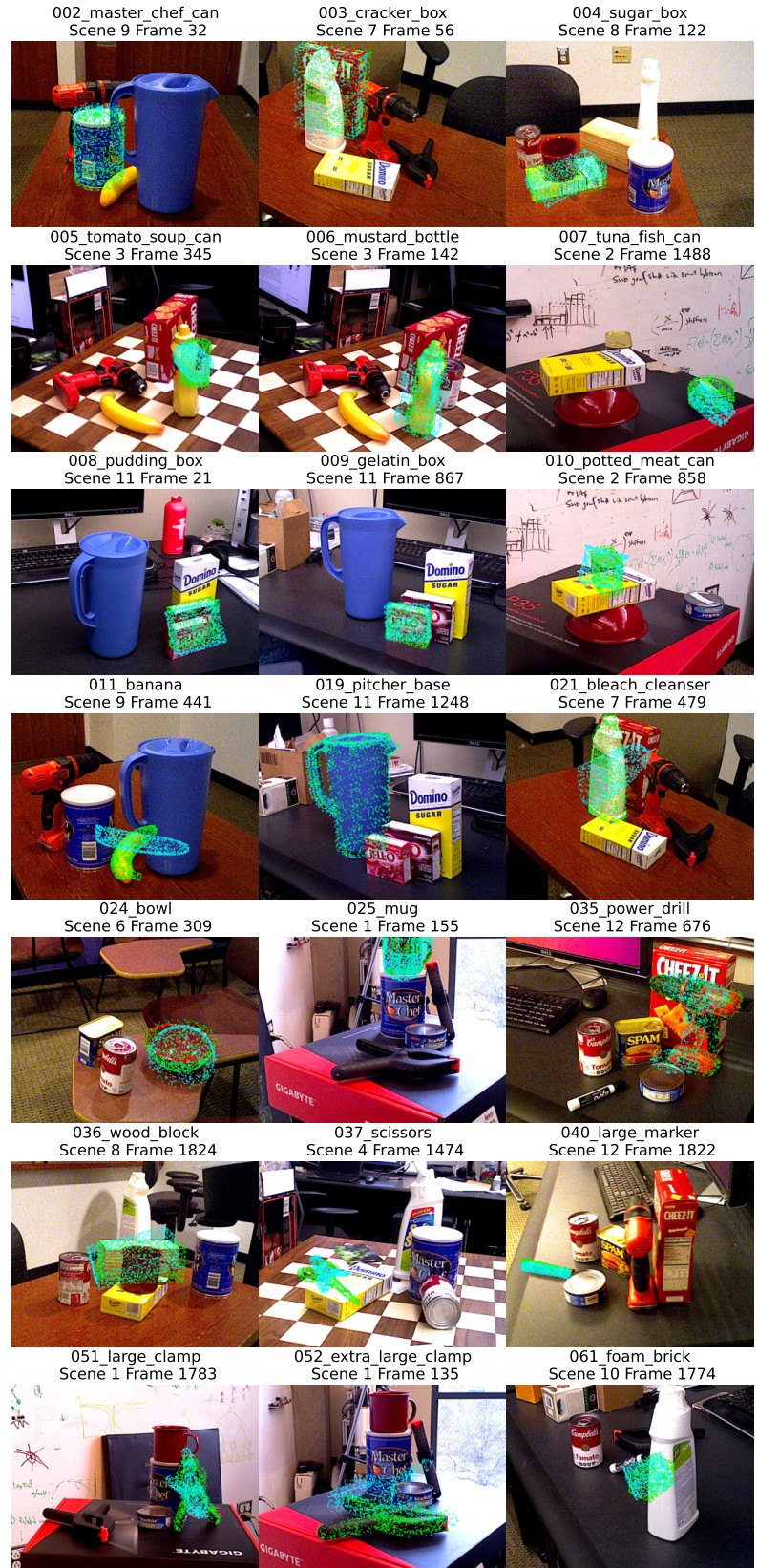

**Figure 5:** YCB frames for each object, overlayed with pose estimates of DenseFusion (cyan) and 3DP3 (green), where there is a large performance difference between the two methods.

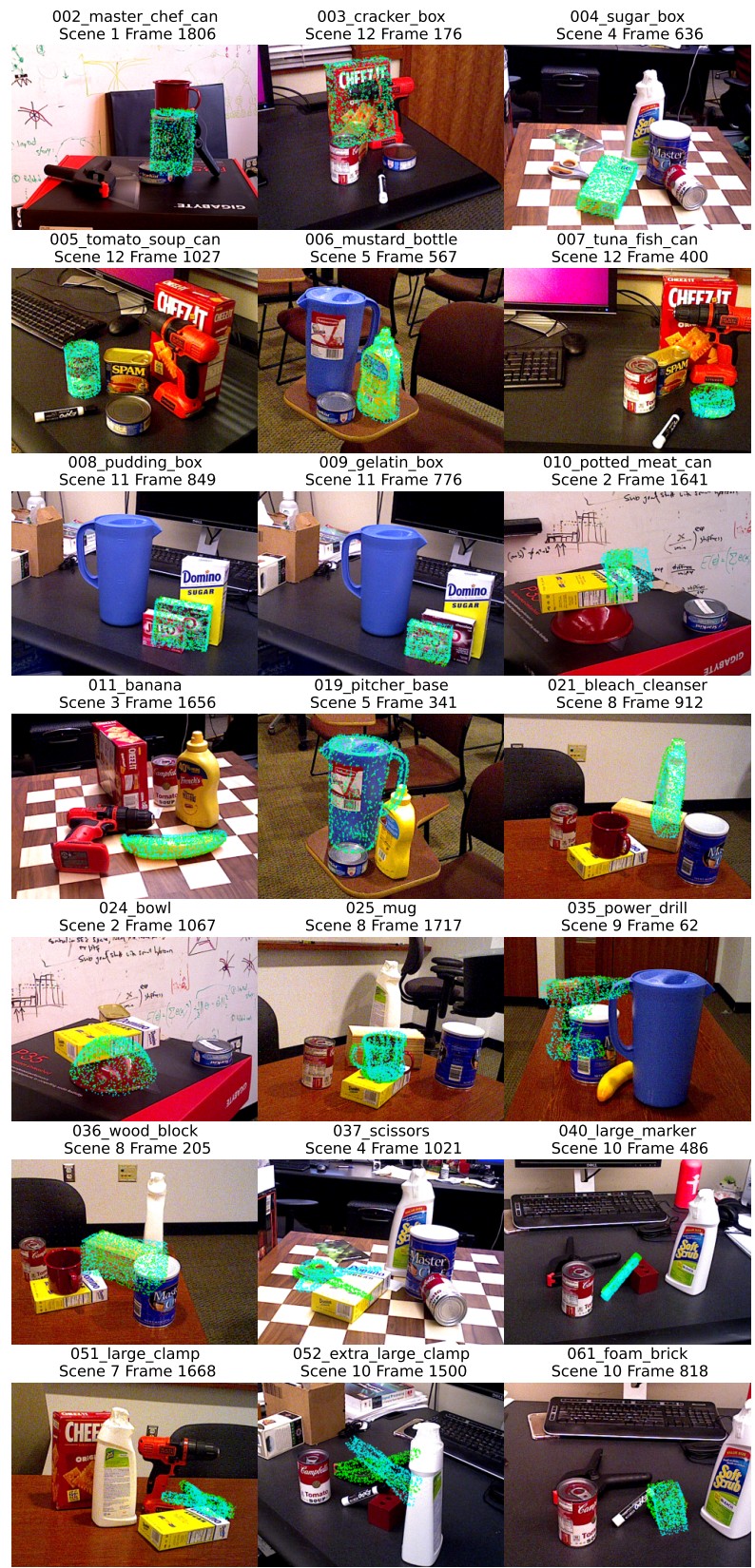

**Figure 6:** YCB frames for each object, overlayed with pose estimates of DenseFusion (cyan) and 3DP3 (green), where there is almost no performance difference between the two methods.

# I  Distilling shape distributions via variational inference

Recall that:

$$p'(\mathbf{O}|\mathbf{I}_{1:T}) = \sum_{k=1}^{K} w_k p'(\mathbf{O}|\mathbf{M}^{(k)}, \mathbf{x}_{1:T}^{(k)}, \mathbf{I}_{1:T}) \tag{1}$$

Consider the following variational family:

$$q_\varphi(\mathbf{O}) := \prod_{i\in[h]} \prod_{j\in[w]} \prod_{\ell\in[l]} \varphi_{ij\ell}^{O_{ij\ell}} \cdot (1 - \varphi_{ij\ell})^{(1-O_{ij\ell})} \tag{2}$$

where each $0 \leq \varphi_{ij\ell} \leq 1$ can be interpreted as a per-voxel occupancy probability. Then

$$\mathrm{KL}(p'(\mathbf{O}|\mathbf{I}_{1:T})||q_\varphi(\mathbf{O})) = \mathbb{E}_{\mathbf{O}\sim p'(\cdot|\mathbf{I}_{1:T})} \left[ \log \frac{p'(\mathbf{O}|\mathbf{I}_{1:T})}{q_\varphi(\mathbf{O})} \right]$$

Note that minimizing this KL divergence with respect to $\varphi$ is equivalent to maximizing the following quantity:

$$\mathbb{E}_{\mathbf{O}\sim p'(\cdot|\mathbf{I}_{1:T})} \left[ \log q_\varphi(\mathbf{O}) \right]$$

$$= \sum_{k=1}^{K} w_k \mathbb{E}_{\mathbf{O}\sim p'(\cdot|\mathbf{M}^{(k)}, \mathbf{x}_{1:T}^{(k)}, \mathbf{I}_{1:T})} \left[ \log q_\varphi(\mathbf{O}) \right]$$

$$= \sum_{k=1}^{K} w_k \mathbb{E}_{\mathbf{O}\sim p'(\cdot|\mathbf{M}^{(k)}, \mathbf{x}_{1:T}^{(k)}, \mathbf{I}_{1:T})} \left[ \sum_{i\in[h]} \sum_{j\in[w]} \sum_{\ell\in[l]} \log q_\varphi(O_{ij\ell}) \right]$$

$$= \sum_{k=1}^{K} w_k \mathbb{E}_{\mathbf{O}\sim p'(\cdot|\mathbf{M}^{(k)}, \mathbf{x}_{1:T}^{(k)}, \mathbf{I}_{1:T})} \left[ \sum_{i\in[h]} \sum_{j\in[w]} \sum_{\ell\in[l]} O_{ij\ell} \log \varphi_{ij\ell} + (1 - O_{ij\ell}) \log(1 - \varphi_{ij\ell}) \right]$$

$$= \sum_{i\in[h]} \sum_{j\in[w]} \sum_{\ell\in[l]} \sum_{k=1}^{K} w_k p'(O_{ij\ell} = 1|\mathbf{M}^{(k)}, \mathbf{x}_{1:T}^{(k)}, \mathbf{I}_{1:T}) \log \varphi_{ij\ell} + p'(O_{ij\ell} = 0|\mathbf{M}^{(k)}, \mathbf{x}_{1:T}^{(k)}, \mathbf{I}_{1:T}) \log(1 - \varphi_{ij\ell})$$

The optimization decomposes into separate problems for each $\varphi_{ij\ell}$, with global optimum:

$$\varphi_{ij\ell}^* = \sum_{k=1}^{K} w_k p'(O_{ij\ell} = 1|\mathbf{M}^{(k)}, \mathbf{x}_{1:T}^{(k)}, \mathbf{I}_{1:T})$$

# J  Probabilistic SLAM using Sequential Monte Carlo

To infer the camera poses $\mathbf{x}_{1:T}$ corresponding to the sequence of $T$ depth images, we implemented a probabilistic SLAM using Sequential Monte Carlo. We assume the depth images contain background, which can be mapped and used for localization between frames. (In our data, the object is placed in the center of a rectangular room with a floor, ceiling, and four walls.) We also assume that the camera is the same distance away from the object in all $T$ images. Finally, in order to ensure a reference frame match between the learned object model and ground truth model (such that at test time, object pose estimates of our system can be compared to the ground truth object poses), we provide the initial pose of the object in the camera frame.

To perform SLAM, we initialize a set of $K$ particles with the observation, camera pose, and implied map at $t = 1$. Then, we enumerate over the position and viewing angle at $t = 2$, given the map at $t = 1$, observation at $t = 2$, and a prior on the camera pose conditioned on the pose at $t = 1$, and compute scores for each pose. We then construct a Gaussian mixture proposal where each component is centered on a different pose and has weight corresponding to the normalized score computed by the enumeration. We step the particles forward to $t = 2$ with this proposal distribution. Then, for each of the particles, we update the map given the observations and inferred poses. We repeat this for all $T$ timesteps, at which point we have $K$ particles with inferred camera poses for each of the $T$ timesteps.

The depth image likelihood $p'(\mathbf{I}_t|\mathbf{O}, \mathbf{M}, \mathbf{x}_t) = \delta_{d(\mathbf{O}, \mathbf{M}, \mathbf{x}_t)}(\mathbf{I}_t)$ where $d$ is a depth rendering function.

## K  Pose initialization for Scene Graph Inference

In the first stage of our scene graph inference algorithm, we obtain initial estimates of the 6DoF object poses via maximum-a-posteriori (MAP) inference in a restricted variant of our model that assumes no edges between objects in the scene graph. We maintain a set of particles with each particle assigned to a different object, and we have at least one particle assigned to each object. Then, for each particle we apply Metropolis-Hastings (MH) kernels to pose of the object that the particle is assigned to. After applying these MH kernels, we resample the set of particles using their normalized weights. We repeat this process of applying the MH kernels and resampling for a fixed number of iterations (proportional to the number of objects). We construct the MH kernels for each object by using spatial clustering and iterative closest point (ICP) to compute a set of "high-quality" poses for each object type given the observed scene. We first apply DBSCAN to the set of points that are unexplained by the current hypothesized scene. Then we create a set of initial object pose hypotheses with translation selected from the $C$ cluster centers output by DBSCAN and orientation selected from the set of 24 nominal orientations, for a total of $24 \cdot C$ poses. (The 24 orientations are the rotational symmetries of a cube.). Next, we refine these initial pose estimates using ICP. The ICP does not use the full object model, but rather renders the object at the hypothesized pose and computes the corresponding point cloud. We score the resulting pose estimates under the generative model and use the normalized weights to construct a mixture proposal that serves as the MH kernel.

In addition to the above MH kernel, we also experimented with kernels based on Boltzmann proposals where the Hamiltonian is determined by performing a 3D convolution of a mask with the observed point cloud. Such proposals can potentially be used as "compiled detectors" of the object models $\mathbf{O}_{1:M}$, enabling us to perform online object learning and scene parsing. This class of proposals takes the following general form:

1. Discretize the observation into a 3D grid $\Gamma$.

2. Given the object model $\mathbf{O}$, create $k$ convolutional masks to be convolved with the grid. Each mask is meant to detect $\mathbf{O}$ at a certain orientation. The candidate orientations are obtained from an appropriately fine geodesic grid on a sphere.

3. Slide each mask over $\Gamma$ and calculate the convolution of the mask and $\Gamma$.

4. Fix $\beta > 0$, and propose a pose from a Boltzmann distribution with temperature $\beta$, where the Hamiltonian of each pose is given by the convolution of its associated mask with $\Gamma$.

We tried multiple approaches for deriving convolutional masks from objects models. Maximally informative and maximally correlated masks require us to solve ill-posed optimization problems. Small windows sampled from the object model are not informative. These masks can give good proposals when combined with expensive ensembling and outlier detection, but they are unsuitable for online settings. Our best results come from globally-sparse, locally-dense [11], randomly selected masks. These masks are computationally efficient to apply and give results that are qualitatively comparable to the ICP-based kernel, but the sampling distribution of the masks have high-variance. In future work, we plan to further investigate this class of proposals.

## L  Parsing scenes with fully occluded objects and number uncertainty

Consider the setting when the number of objects in the scene ($N$) is unknown a-priori. Possibilities for prior probability distributions on $N$ ($p(N)$) include (i) an a-priori known number of objects $N_0$ (used in the experiments in Section 6), (ii) $\text{Binomial}(N_0, p_{\text{present}})$, which is induced by a prior belief that each of $N_0$ objects is present with probability $p_{\text{present}}$ (used in experiments described in Supplement L), and (iii) $\text{Poisson}(\lambda)$, which places no a-priori upper bound on the number of objects.

In this section, we apply our framework to do probabilistic inference about the 6DoF pose and presence or absence of a fully occluded object, and investigate the dynamics of these inferences as we vary the fraction of the volume in the scene that is occluded.

Suppose a robot is tasked with assembling a piece of furniture, performing maintenance on a vehicle, or retrieving something from the kitchen. In each of these cases, the robot has a strong prior expectation that some object (e.g. a tool, component, or kitchen item) is present in the environment. However, in complex cluttered real-world environments the target object is likely to be fully occluded

from the robot's view. That target object may even even be absent, especially in human-robot interactive task situations (e.g. the component or item is missing or misplaced). To perform rationally in such situations, the robot will need to generate possible poses of the object that are concordant with its *absence* from its visual field. Also, the robot must consider the possibility that the object is indeed not present, by weighing the lack of observed presence of the object against the prior expectations.

**Prior** Consider a scenario where there are $N_0 = M$ unique objects that may or may not be present in a scene ($N_0$ denotes the total number of object instances, and $M$ denote the number of object types). Suppose that the prior probability that the object of type $m$ is present with probability $p_{\text{pres}}^{(m)}$ for $m \in \{1, \ldots, M\}$. Then, the prior $p(N, \mathbf{c})$ is:

$$p(N, \mathbf{c}) = \begin{cases} \frac{1}{N!} \prod_{m=1}^{M} p_{\text{pres}}^{(m)} {}^{\mathbf{1}[m \in \mathbf{c}]} (1 - p_{\text{pres}}^{(m)})^{\mathbf{1}[m \notin \mathbf{c}]} & \text{if } |\mathbf{c}| = N \text{ and } \sum_{i=1}^{N} \mathbf{1}[c_i = m] \leq 1 \, \forall m \\ 0 & \text{otherwise} \end{cases}$$

(3)

In the special case when $p_{\text{pres}}^{(m)}$ is the same for all $m$ (this is the case in our experiments below), we can write the marginal distribution $p(N)$ and conditional distribution $p(\mathbf{c}|N)$ as:

$$N \sim \text{Binomial}(M = N_0, p_{\text{pres}})$$

(4)

and

$$p(\mathbf{c}|N) = \begin{cases} \frac{(M-N)!}{M!} & \text{if } |\mathbf{c}| = N \text{ and } \sum_{i=1}^{N} \mathbf{1}[c_i = m] \leq 1 \, \forall m \\ 0 & \text{otherwise} \end{cases}$$

(5)

For each possible $(N, \mathbf{c})$, we fix the scene graph $G$ to be the graph $G_0(N)$ on $N$ vertices that has with no object-object edges:

$$p(G|N) = \begin{cases} 1 & \text{if } G = G_0(N) \\ 0 & \text{otherwise} \end{cases}$$

(6)

That is, each object $v$ has an independent 6DoF pose $\mathbf{x}_v$. The prior distribution on the orientation component used the uniform distribution on an Euler angle parametrization, and the prior on the translation component (i.e. the location of the object) the uniform distribution on a cuboid volume representing the extent of the scene.

**Likelihood** Instead of the likelihood on point clouds used in Section 3.3, here we use an alternative likelihood based on (i) rendering a depth image $\tilde{\mathbf{I}}(\mathbf{O}_{1:M}, \mathbf{c}, G, \boldsymbol{\theta})$ and then (ii) adding noise to generate an observed depth image $\mathbf{I}$. The likelihood is a per-pixel mixture between a uniform distribution on the range of possible depth values, and a normal distribution with fixed variance $\sigma^2$:

$$p(\mathbf{I}|x) = \prod_i \left( 0.1 \cdot \frac{1}{D} + 0.9 \cdot \mathcal{N}(I_i; \tilde{I}_i, \sigma)) \right)$$

where $i$ indexes pixels of the depth image. Pixels whose ray does not intersect an object are assigned the maximum depth value $D$. A similar likelihood function on depth images was used in [10].

**MCMC inference algorithm** We use a Markov chain Monte Carlo (MCMC) inference algorithm that cycles through each object type $m \in \{1, \ldots, M\}$, and applies several types of MCMC moves for each type, based on the following proposals: (i) involutive MCMC kernels that switch an object type from being absent to being present and vice versa (proposing its pose from the prior) , (ii) Metropolis–Hastings kernels that propose the translational components of the pose $\mathbf{x}_v$ for each object from the prior, (iii) Metropolis–Hastings kernels that propose the rotational component of the pose for each object from the prior, and (iv) coordinate-wise random-walk proposals to each of the 6 dimensions of the pose of each object (the 3 coordinates of its location and its Euler angles). We initialize the Markov chain with a sample from the prior distribution.

**Inferring the 6DoF pose of a fully occluded object** We first investigated inference about the 6DoF pose of an object (the mug from the YCB object set [3]) that is assumed to be in a volume in front of the camera, but that is not visible. This scenario arises when searching for a component or tool that is expected to be in the environment. Narrowing down where the object could be, based on observing where it is not, is important for efficiently planning and acting to obtain more information

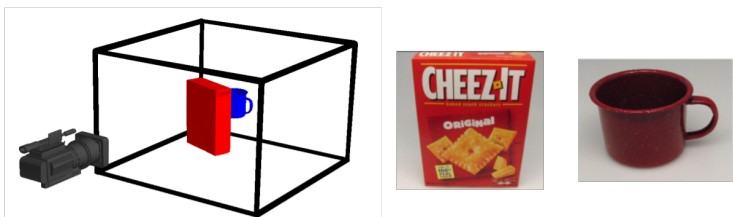

**(a)** The scenario (left). A depth camera is viewing a scene that may or may not contain a cracker box (middle) and a mug (right). We perform Bayesian inference on the existence and contingently, their 6DoF poses within the scene, of both objects. Only the cracker box is visible in the observed depth images (see below).

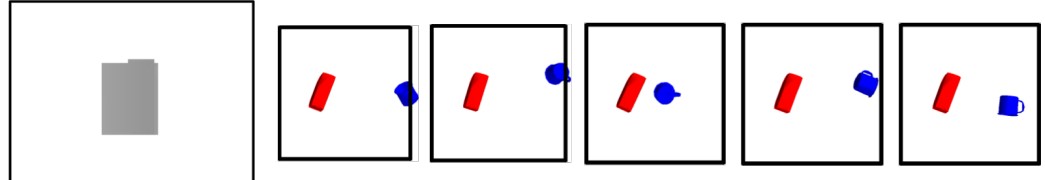

**(b)** Observed depth image and posterior samples where the existence of both the box and mug are assumed.

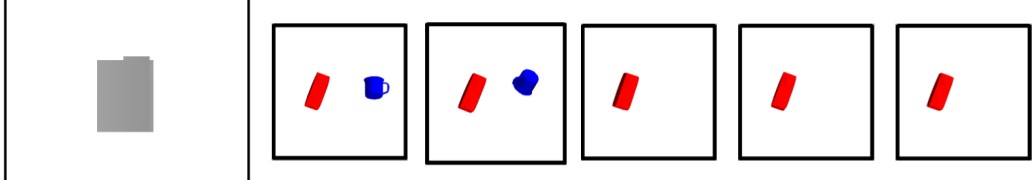

**(c)** Observed depth image and posterior samples where the existences of each object have prior probability 0.90. The posterior probabilities of existence for the mug and box are 0.37 and 1.0, respectively.

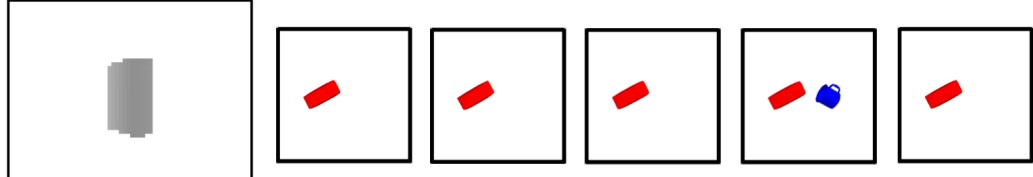

**(d)** Observed depth image and posterior samples where the existences of each object have prior probability 0.90. Note that the observed image has the box angled so that it occupies less of the field of view than in (c). The posterior probabilities of existence for the mug and box are 0.33 and 1.0, respectively.

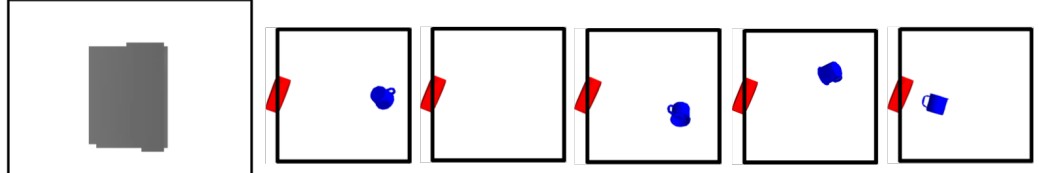

**(e)** Observed depth image and posterior samples where the existences of each object have prior probability 0.90. Note that the observed image has the box closer to the camera, so that it occupies more of the field of view than in (c). The posterior probabilities of existence for the mug and box are 0.63 and 1.0, respectively.

**Figure 7:** Inferring the 6DoF pose and existence of multiple objects from depth images using MCMC in a generative model. Several scenarios are shown, with five approximate posterior samples from each. To estimate posterior probabilities of object existence, 20 posterior samples were used. The *lack of percept* of the mug in the visual field (i) reduces the posterior probability of its presence, but also (ii) informs the distribution on its 6DoF pose, if it is present. Note that as the fraction of volume in the scene that is occluded by the box decreases, the posterior probability that mug is present decreases.

and retrieve the object. In order for inference to be coherent, the lack of the object's visible presence must be explained away by the occluding presence of another object. Therefore, we also assume the presence of another object (the cracker box). The results (Figure 7(b)) show that the algorithm successfully infer a variety of 6DoF poses of the mug in which it lies behind the cracker box.

**Jointly inferring the existence and poses of multiple objects**  If an object is not visible, we may conclude that it is not present in the scene. The degree of belief in the presence of an object that is not visible depends on the degree of prior belief in its presence and the volume of possible states in which the object is fully occluded. If there are no other objects in the scene, then intuitively, there is nowhere the object could be hiding in the scene, so it must be absent. If there are other objects in the scene, then the pose of these other objects interacts with the object's existence, in our beliefs. To investigate the interaction between the beliefs about the poses and existence of multiple objects, we generated synthetic depth data for several scenarios (Figure 7(c-e)). In all scenarios, the prior probabilities that the box and mug are present are both 0.9. In the first scenario (Figure 7(c)) the box is oriented so a wide face is facing the camera. We correctly infer the existence of the box with high confidence (1.0 posterior probability) and we assign 0.37 posterior probability to the existence of the mug. In the second scenario (Figure 7(d)) the box is rotated so that it occupies less of the camera's field of view. As expected, this causes the posterior probability of the mug's existence to decrease to 0.31. In the third scenario (Figure 7(e)), we move the box closer to the camera so that it occupies more of the field of view. The posterior probability of the mug's presence then increases to 0.63. These experiments illustrate the dependence between one object's pose and another object's existence, which is a consequence of occlusion.

## M   Experiment Details

The deep learning baseline experiments were run on a 3.70GHz Intel i7 processor with 64GB RAM and a Nvidia GeForce GTX 1080Ti GPU. All other experiments were run a 2.40GHz Intel i9 processor with 32GB RAM and a Nvidia GeForce GTX 1650 Mobile GPU. Our model is implemented using Julia in Gen, a probabilistic programming system [6].

## N   Pseudomarginal shape inference

We use MH and involutive MCMC kernels that are stationary with respect to a target distribution on an extended state space that includes auxiliary variables $\mathbf{O}^{(i)}$ for $i = 1, \ldots, R$ ($R$ copies of all object shape models) by replacing the likelihood $p(\mathbf{Y}|N, \mathbf{c}, G', \boldsymbol{\theta}')$ for each proposed state in the acceptance probability with the following unbiased estimate obtained by sampling object models from the prior:

$$\frac{1}{R} \sum_{i=1}^{R} p(\mathbf{Y}|\mathbf{O}_1^{(i)}, \ldots, \mathbf{O}_M^{(i)}, N, \mathbf{c}, G, \boldsymbol{\theta}) \text{ where } \mathbf{O}_c^{(i)} \overset{\text{i.i.d.}}{\sim} p(\cdot) \tag{7}$$

and replacing the likelihood in the denominator of the acceptance ratios with the unbiased estimate computed by the last accepted proposal. This is an instance of the pseudomarginal MCMC [1] framework. For example, for the scene graph involutive MCMC kernel described in Section O, each factor of the form:

$$\frac{p(\mathbf{Y}|N, \mathbf{c}, G', \boldsymbol{\theta}')}{p(\mathbf{Y}|N, \mathbf{c}, G, \boldsymbol{\theta})} \tag{8}$$

is replaced with a factor:

$$\frac{\frac{1}{R} \sum_{i=1}^{R} p(\mathbf{Y}|\mathbf{O}_1^{(i)'}, \ldots, \mathbf{O}_M^{(i)'}, N, \mathbf{c}, \mathcal{G})}{\frac{1}{R} \sum_{i=1}^{R} p(\mathbf{Y}|\mathbf{O}_1^{(i)}, \ldots, \mathbf{O}_M^{(i)}, N, \mathbf{c}, \mathcal{G})} \tag{9}$$

where the $\mathbf{O}_j^{(i)}$ are the shape models that were sampled when proposing the last state that was accepted, and where the $\mathbf{O}_j^{(i)'}$ are the shape models that are sampled during the current proposal. Note that the old sampled shape models $\mathbf{O}_j^{(i)}$ themselves do not need to be persisted across steps of the Markov chain—only the denominator in the expression above needs to be stored. The resulting

moves can be interpreted as MH (or involutive MCMC) moves on an extended state space that includes additional auxiliary random variables $\mathbf{O}_{1:M}^{(1)}, \ldots, \mathbf{O}_{1:M}^{(R)}$. The moves are stationary with respect to the following target distribution on the extended state space:

$$p(G, \boldsymbol{\theta}|N, \mathbf{c}, \mathbf{Y}) \frac{1}{R} \sum_{i=1}^{R} p(\mathbf{O}_{1:M}^{(i)}|N, \mathbf{c}, G, \boldsymbol{\theta}, \mathbf{Y}) \prod_{j \neq i} p(\mathbf{O}_{1:M}^{(j)}) \tag{10}$$

of which the marginal on $(G, \boldsymbol{\theta})$ is the original target distribution $p(G, \boldsymbol{\theta}|N, \mathbf{c}, \mathbf{Y})$.

## O  Involutive MCMC kernel on scene graph structure and parameters

This section gives details of the involutive MCMC kernel on scene graphs introduced in Section 5.

### O.1  Notation for coordinate projections

In several places below, we define sets of tuples using set-builder notation such as

$$X := \{(x, y, z) \mid \text{some condition on } x, y, z\}.$$

In such a case, we may also define coordinate projections that get their names from the formal variables ("$x$," "$y$," "$z$") used in the set-builder expression. Our convention is to denote these coordinate projections by the name $\text{proj}_\bullet$, where $\bullet$ is either a variable name, e.g.,

$$\text{proj}_x(x, y, z) := x$$

or a comma-separated list of variable names, e.g.,

$$\text{proj}_{y,z}(x, y, z) := (y, z).$$

This definition depends not just on the set $X$, but on the notation used to define it; thus, in the exposition below, we explicitly declare each time we define a function $\text{proj}_\bullet$ using the above convention. Note that the name $\text{proj}_x$ is to be taken as a single unit, i.e., $x$ does not have independent meaning; in particular, if there is also a variable $x$ in the scope of discourse, the name $\text{proj}_x$ does not have anything to do with that variable's value.

### O.2  The number of scene graph structures on a fixed set of objects

**Proposition O.2.1.** *For a given set of $N$ objects, the number of possible scene graph structures is $(N+1)^{N-1}$. (Here the root node $r$ is not considered an object.)*

*Proof.* Let $V'$ be a set of $N$ objects, and let $V := V' \cup \{r\}$. For a given *undirected* tree $\widetilde{G}$ on vertices $V$, there is a unique way to assign edge directions to $\widetilde{G}$ to turn it into a directed tree rooted at $r$. This gives a one-to-one correspondence

$$\text{directed trees on } V \text{ rooted at } r \quad \longleftrightarrow \quad \text{undirected trees on } V.$$

By Cayley [4], the number of undirected trees on $V$ is $(N+1)^{N-1}$. $\qquad \square$

### O.3  An involutive MCMC kernel on scene graph structure only

For some set $V$ of vertices and a root vertex $r \in V$, let $\mathbf{G}(V)$ denote the set of directed trees over vertices $V$ rooted at $r$. For each $G = (V, E) \in \mathbf{G}(V)$ and each $v \in V \setminus \{r\}$, let $S(G, v) \subset V$ denote the vertices of the subtree rooted at $v$, i.e., the set containing $v$ and its descendants. Let

$$T(G) := \{(v, u) \subset V \times V : v \neq r, u \notin S(G, v)\}. \tag{11}$$

That is, $T(G)$ contains a every pair of vertices $(v, u)$ such that $v$ is not the root note, and $u$ is not a descendant of $v$. (Intuitively, we can think of $T(G)$ as the set of pairs of vertices $(v, u)$ such that it is possible to sever the subtree rooted at $v$ and re-attach that subtree as a child of $u$.) Next, let

$$U(V) := \{(G, v, u) : G \in \mathbf{G}(V), (v, u) \in T(G)\} \tag{12}$$

and equip $U(V)$ with coordinate projections $\mathrm{proj}_G$, $\mathrm{proj}_v$, etc. as in Section O.1. (Intuitively, we can think of the triples $(G, v, u) \in U(V)$ as denoting a graph $G$, a choice of vertex $v$ at which to sever, and a choice of vertex $u$ at which to graft.) Finally, define the function

$$g : U(V) \to U(V)$$

by $g(G, v, u) = (G', v, u')$, where (i) $u'$ is the parent of $v$ in $G$, and (ii) $G'$ is the graph obtained from $G$ by removing the edge $(u', v)$ and adding the edge $(u, v)$. (By Lemma O.7.1, $G'$ is a tree, so $(G', v, u') \in U(V)$.)

**Proposition O.3.1.** *The function $g : U(V) \to U(V)$ is an involution.*

*Proof.* Let $(G', v', u') := g(G, v, u)$; let $(G'', v'', u'') := g(G', v', u')$; and let $E$, $E'$ and $E''$ be the edge sets of $G$, $G'$ and $G''$ respectively. Then, by the definition of $g$, we have $v'' = v' = v$. Also, because $G'$ is a tree that contains the edge $(u, v)$, it follows that $u$ is the parent of $v$ in $G'$. Thus $u'' = u$, since $u''$ is also (by definition) the parent of $v$ in $G'$. Next, by the definition of $g$, we have $E' = (E \setminus \{(u', v)\}) \cup \{(u, v)\}$ and

$$E'' = (E' \setminus \{(u, v)\}) \cup \{(u, v)\} = (E \setminus \{(u', v)\}) \cup \{(u', v)\} = E$$

(here we are using the fact that $(u', v) \in E$, which holds by the definition of $g$). Thus $G'' = G$, so $g(g(G, v, u)) = (G, v, u)$. $\qquad \square$

**Proposition O.3.2.** *Let $G, G' \in \mathbf{G}(V)$. Then there exists a sequence of triples*

$$(G_0, v_0, u_0), \ldots, (G_k, v_k, u_k)$$

*satisfying all of the following:*

*(i) $(G_i, v_i, u_i) \in U(V)$ for all[1] $i = 0, \ldots, k - 1$*
*(ii) $G_0 = G$*
*(iii) $G_k = G'$*
*(iv) for each $i < k$ we have $G_{i+1} = \mathrm{proj}_G(g(G_i, v_i, u_i))$.*

*Proof.* Let $G^\circ$ denote the scene structure that has no relations between objects; that is,

$$G^\circ := (V, \{(r, v) : v \in V \setminus \{r\}\}).$$

We first prove the result in the case where $G = G^\circ$; then we extend to the general case.

For the case where $G = G^\circ$, the intuition is to work backwards from $G'$ to $G^\circ$ by grafting subtrees onto $r$ until there are no non-singleton subtrees left. Let $G' = (V, E')$, and let $\widetilde{E} := \{(u, v) \in E : v \neq r\}$. We then take[2] $k := |\widetilde{E}| - 1$. We define $v_i$ (from the proposition statement) and $u_i'$ (a variable we're now introducing) by arbitrarily choosing an ordered enumeration of $\widetilde{E}$ and defining the sequence of pairs $(u_0', v_0), \ldots, (u_k', v_k)$ to equal that enumeration. Thus,

$$\widetilde{E} = \{(u_0', v_0), \ldots, (u_k', v_k)\}.$$

Next, we define $G_i$ and $u_i$ simultaneously by backward recursion: we take $G_k := G'$ (and choose $u_k$ arbitrarily; the proposition doesn't actually say anything about $u_k$); and for $0 \leq i < k$, we define $G_i$ and $u_i$ by the equation

$$g(G_{i+1}, v_i, r) = (G_i, v_i, u_i). \tag{13}$$

(To justify the left-hand side being well-defined, we must show $(G_{i+1}, v_i, r) \in U(V)$ for all $i$. By construction, $v_i \neq r$ for all $i$, so $r \notin S(G_{i+1}, v)$; hence $(G_{i+1}, v_i, r) \in U(V)$.) By applying $g$ to both sides of (13), we get (iv). Finally, note that by induction, $G_i$ has exactly $i$ edges whose source node is not $r$. Thus $G_0 = G^\circ$. This completes the proof of the case where $G = G^\circ$.

We now move to the general case, where $G \in \mathbf{G}(V)$ is arbitrary. Using the above special case, let $(G_0 = G^\circ, v_0, u_0), (G_1, v_1, u_1), \ldots, (G_k, v_k, u_k)$ be a sequence satisfying (i)–(iv) and (13) with $G_0 = G^\circ$. Let $(\overline{G}_0, \overline{v}_0, \overline{u}_0), (\overline{G}_1, \overline{v}_1, \overline{u}_1), \ldots, (\overline{G}_k, \overline{v}_k, \overline{u}_{\overline{k}})$ be a sequence satisfying (i)–(iv) and (13)

---

[1]This proposition doesn't say anything about $u_k$ and $v_k$; we leave them in just to simplify the exposition and notation.

[2]Except in the degenerate case where $\widetilde{E}$ is empty. In that case we have $G' = G^\circ$, so we simply take $k := 0$, $G_0 := G^\circ$, and $u_0$ and $v_0$ to be any vertices.

but with $\overline{G}_0 = G^\circ$ and $\overline{G}_{\overline{k}} = G$. Let $\underline{G}_j := \overline{G}_{\overline{k}-j}$ for $0 \le j < \overline{k}$ and $\underline{G}_{\overline{k}} := G^\circ$. Substituting $j := \overline{k} - (i+1)$ into (13) gives

$$g\left(\underline{G}_j,\ \overline{v}_{\overline{k}-j-1},\ r\right) = \left(\underline{G}_{j+1},\ \overline{v}_{\overline{k}-j-1},\ \overline{u}_{\overline{k}-j-1}\right)$$

for each $j = 0, \dots, \overline{k}-1$. Thus, letting $\underline{k} := \overline{k}$ and $\underline{v}_j := \overline{v}_{\overline{k}-j-1}$ and $\underline{u}_j := r$, the sequence

$$(\underline{G}_0, \underline{v}_0, \underline{u}_0), \dots, (\underline{G}_{\underline{k}}, \underline{u}_{\underline{k}}, \underline{v}_{\underline{k}})$$

satisfies (i)–(iv) but with $\underline{G}_0 = G$ and $\underline{G}_{\underline{k}} = G^\circ$. Then, the sequence

$$(\underline{G}_0, \underline{v}_0, \underline{u}_0), \dots, (\underline{G}_{\underline{k}-1}, \underline{u}_{\underline{k}-1}, \underline{v}_{\underline{k}-1}),\ (\underline{G}_{\underline{k}} = G_0 = G^\circ, v_0, u_0),\ (G_1, v_1, u_1), \dots, (G_k, v_k, u_k)$$

satisfies (i)–(iv): the only piece of this claim that wasn't already proved above is (iv) at the concatenation boundary where $G^\circ$ appears; i.e., it remains only to check that

$$G^\circ = \mathrm{proj}_G(g(\underline{G}_{\underline{k}-1}, \underline{v}_{\underline{k}-1}, \underline{u}_{\underline{k}-1})).$$

Unpacking the definitions, this condition is

$$G^\circ = \mathrm{proj}_G(g(\overline{G}_1, \overline{v}_0, r)),$$

and indeed the condition is satisfied, by (13). $\qquad\square$

For some fixed $V$ with $r \in V$ and $N := |V| - 1$, suppose we have in hand (i) a prior probability distribution $p(G)$ on $G \in \mathbf{G}(V)$; (ii) some data $\mathcal{D}$; and (iii) a likelihood function $p(\mathcal{D}|G)$. From these, we can construct an involutive MCMC [5] kernel on the space of graphs ($\mathbf{G}(V)$) by combining the involution $g$ defined above with a family of auxiliary probability distributions $q(v, u; G)$ on $V \times V$ such that $q(v, u; G) > 0$ if and only if $(v, u) \in T(G)$. The kernel takes as input a graph $G$, samples $(v, u) \sim q(\cdot; G)$, computes $(G', v', u') := g(G, v, u)$, and then returns the new graph $G'$ with probability

$$\min\left\{1, \frac{p(G')p(\mathcal{D}|G')q(v', u'; G')}{p(G)p(\mathcal{D}|G)q(v, u; G)}\right\} \tag{14}$$

and otherwise returns the previous graph $G$ (i.e. rejects). Because this kernel satisfies the requirements of involutive MCMC, it is stationary with respect to the following target distribution on $\mathbf{G}(V)$:

$$p(G|\mathcal{D}) = \frac{p(G)p(\mathcal{D}|G)}{\sum_{G' \in \mathbf{G}(V)} p(G')p(\mathcal{D}|G')} \tag{15}$$

Furthermore, if $p(G) > 0$ and $p(\mathcal{D}|G) > 0$ for all $G \in \mathbf{G}(V)$, then the Markov chain generated by repeated application of this kernel converges to the target distribution as the number of steps goes to infinity (the chain is irreducible by Prop. O.3.2 and aperiodic since the proposal has a positive probability of choosing $u$ to be the parent of $v$, and in that case $G' = G$).

## O.4 Transforming between two alternative 6DoF pose parametrizations

Before defining our full involutive MCMC kernel on scene graphs, we define a transformation between continuous parameter spaces that will be used as a building block of the full kernel.

Objects $v$ that are children of the root vertex have their 6DoF pose relative to the world coordinate frame parametrized directly via $\theta_v \in SE(3)$. Recall that the pose of an object $v$ that is child of another object $u'$ is parametrized via the relative pose between a face of $u'$ and a face of $v$. We choose a parametrization for the pose of one face relative to another that makes it natural to express a prior distribution in which: (i) the two faces are nearly in flush contact with high probability, and (ii) we know little about the relative in-plane offset of the two faces and their relative in-plane orientation. A natural parametrization for this prior uses:

- Two dimensions for the in-plane offset ($a \in \mathbb{R}$ and $b \in \mathbb{R}$) with relatively broad priors
- One dimension for the perpendicular offset ($z \in \mathbb{R}$) with a concentrated prior
- An (outward) face normal vector ($\nu \in S^2$) with a prior that concentrates on anti-parallel face normals

- One dimension of in-plane angular rotation ($\varphi \in S^1$) with a uniform prior.

We now define a bijection

$$
\begin{aligned}
\xi : \mathbb{R}^3 \times SO(3) \supseteq \mathbb{R}^3 \times \{\omega \in SO(3) : \omega(0,0,1)^\top \neq (0,0,-1)\} \\
\to \mathbb{R} \times \mathbb{R} \times \mathbb{R} \times (S^2 \setminus \{(0,0,-1)^\top\}) \times S^1 \subseteq \mathbb{R} \times \mathbb{R} \times \mathbb{R} \times S^2 \times S^1. \quad (16)
\end{aligned}
$$

Note that $\xi$ is a.e. bijective on the supersets as well, in the sense that in (16), $\xi$ is a bijection between the subsets, and each of the subsets has a complement of measure zero in its superset. In $\xi$, the first three coordinates are copied directly and the orientations are transformed as follows.

In the first parameter space, $SO(3)$, orientations are represented "directly," as the linear transformation that carries the parent coordinate frame to the child coordinate frame (translated to have the same origin). The base measure on $SO(3)$ is the Haar measure.

In the second parameter space, $S^2 \times S^1$, orientations are represented in Hopf coordinates [14]: intuitively, these coordinates characterize a rotation by where it carries the north pole $(0,0,1)^\top$ (we call this $\eta \in S^2$) and how much planar rotation it does after carrying the north pole to $\eta$ (we call this $\varphi \in S^1$). However, there is no globally consistent (i.e., jointly continuous in $\eta$ and $\varphi$) way to choose where the rotation corresponding to $\varphi$ "starts" (i.e. which orientations have $\varphi = 0$)—formally, the fiber bundle induced by the Hopf fibration is not a trivial bundle. But, it is possible to make these choices consistently on an open subset of $S^2 \times S^1$ whose complement has measure zero, as we do in Section O.6.2. In particular, provided that $\eta$ is not the south pole $(0,0,-1)$, there is a unique unit quaternion $(w,x,y,z) \in S^3$ that satisfies $\mathrm{spin}(w,x,y,z)(0,0,1)^\top = \eta$ and has minimal geodesic distance to the identity (where $\mathrm{spin} : S^3 \to SO(3)$ is the usual covering map, described in Section O.6.1). Then, $\mathrm{spin}(w,x,y,z) \in SO(3)$ is the rotation we take to correspond to $\eta = \eta$, $\varphi = 0$. The resulting map $(S^2 \setminus \{(0,0,-1)^\top\}) \times S^1 \to SO(3)$ is smooth; in Section O.6.2 we compute the mapping explicitly in terms of coordinates.

## O.5 Full involutive MCMC kernel on scene graphs

Our full involutive MCMC kernel on scene graphs (including their parameters $\boldsymbol{\theta}$) is based on an extension of the involution $g$ defined above. We first define the latent space of pairs $(G, \boldsymbol{\theta})$, as:

$$
X := \bigsqcup_{G \in \mathbf{G}(V)} \left( \left( \underset{\substack{v \in V \setminus \{r\} \\ (r,v) \in E}}{\times} SE(3) \right) \times \left( \underset{\substack{v \in V \setminus \{r\} \\ (r,v) \notin E}}{\times} (F \times F \times \mathbb{R} \times \mathbb{R} \times \mathbb{R} \times S^2 \times S^1) \right) \right) \quad (17)
$$

($\sqcup$ denotes disjoint union), and we endow this set with a reference measure $\mu_P$ formed by sums (over the disjoint union) of product measures that are composed from: the Lebesgue measure on $\mathbb{R}$, the Haar measure on $SO(3)$, the uniform (spherical) measures on $S^2$ and $S^1$, and the counting measure on $F \times F$. Then, we define the space of auxiliary variables as:

$$
Y := \{(v,r) : v \in V \setminus \{r\}\} \sqcup \{(v,u,f,f') : v,u \in V \setminus \{r\} \text{ and } f, f' \in F\} \quad (18)
$$

with the reference measure $\mu_Q$ being the counting measure. We define an auxiliary probability distribution $q(y; G, \boldsymbol{\theta})$ such that

$$
\begin{array}{llll}
q(v, r; G, \boldsymbol{\theta}) & > 0 & \text{for all } v \in V \setminus \{r\} \\
q(v, u, f, f'; G, \boldsymbol{\theta}) & > 0 & \text{for all } (v,u) \in T(G) \text{ and all } f, f' \in F \\
q(y; G, \boldsymbol{\theta}) & = 0 & \text{otherwise.}
\end{array} \quad (19)
$$

The extended state space of the involutive MCMC kernel is then

$$
Z := \{(x,y) \in X \times Y : p(x)q(y;x) > 0\}. \quad (20)
$$

We construct an involution $h$ on the space $Z$ using the graph involution $g$ defined above as a building block. In particular, $Z$ consists of tuples of four forms, and we define the involution $h$ piecewise depending on which of these four forms the input has: we take

$$
Z = Z_1 \sqcup Z_2 \sqcup Z_3 \sqcup Z_4
$$

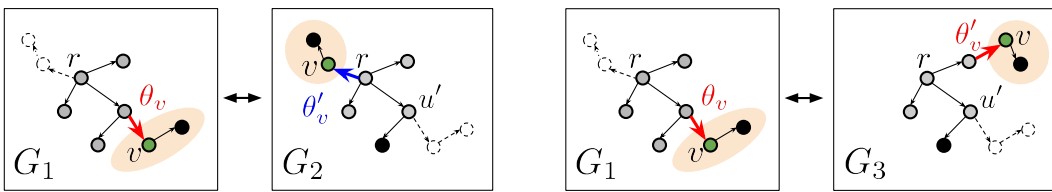

**Figure 8:** A subset of the possible transition types for our involutive MCMC kernel on scene graphs. Left: 'contact to floating' (forwards) and 'floating to contact' (backwards). Right: 'contact to contact' (forwards) and 'contact to contact' (backwards). The vertex $v$ is the chosen 'sever' vertex, and its subtree $S(G, v)$ is shaded. Parameters $\theta_v \in SE(3)$, which are the independent 6DoF pose of an object relative to the world coordinate frame, are shown in blue; and parameters $\theta_v \in F \times F \times \mathbb{R} \times \mathbb{R} \times \mathbb{R} \times S^2 \times S^1$, which parametrize the pose of an object relative to another object (specifically the relative pose between two faces of the two objects) are shown in red.

where the definitions of $Z_1, Z_2, Z_3, Z_4$, and the parametric form that $\theta_v$ takes on each component, are as follows:

$$Z_1 := \{(G, v, r, \boldsymbol{\theta}) : (r, v) \in E\} \qquad \theta_v \in SE(3)$$
$$Z_2 := \{(G, v, r, \boldsymbol{\theta}) : (r, v) \notin E\} \qquad \theta_v \in F \times F \times \mathbb{R} \times \mathbb{R} \times \mathbb{R} \times S^2 \times S^1$$
$$Z_3 := \{(G, v, u, \boldsymbol{\theta}, f, f') : u \neq r, (r, v) \in E\} \qquad \theta_v \in SE(3)$$
$$Z_4 := \{(G, v, u, \boldsymbol{\theta}, f, f') : u \neq r, (r, v) \notin E\} \qquad \theta_v \in F \times F \times \mathbb{R} \times \mathbb{R} \times \mathbb{R} \times S^2 \times S^1.$$

We define coordinate projections $\text{proj}_\bullet$ on $Z_1$ and $Z_2$ (for the free parameters $G, v, \boldsymbol{\theta}$), and on $Z_3$ and $Z_4$ (for the free parameters $G, u, \boldsymbol{\theta}, f, f'$), as in Section O.1. Also, for notational convenience below, we extend $\text{proj}_u$ to $Z_1$ and $Z_2$ by defining $\text{proj}_u(G, v, r, \boldsymbol{\theta}) := r$; i.e., $\text{proj}_u$ is constant on $Z_1 \sqcup Z_2$ with value $r$.

Then, $h$ is defined piecewise as follows:

$$h(z) = \begin{cases} h_{\text{f}\to\text{f}}(z) & \text{if } z \in Z_1 \quad \text{(floating to floating)} \\ h_{\text{c}\to\text{f}}(z) & \text{if } z \in Z_2 \quad \text{(contact to floating)} \\ h_{\text{f}\to\text{c}}(z) & \text{if } z \in Z_3 \quad \text{(floating to contact)} \\ h_{\text{c}\to\text{c}}(z) & \text{if } z \in Z_4 \quad \text{(contact to contact).} \end{cases} \tag{21}$$

We next define the function $h_{\bullet\to\bullet}$ corresponding to each of these four components, and give the acceptance probability in each case (the acceptance probabilities will be derived later in this section).

**Floating to floating** This transition makes no change to the structure $G$ or the parameters $\boldsymbol{\theta}$:

$$h_{\text{f}\to\text{f}}(G, v, r, \boldsymbol{\theta}) := (G, v, r, \boldsymbol{\theta}) \quad \text{(no change)} \tag{22}$$

**Contact to floating** This transition severs the edge from the parent of $v$ in $G$ (another object) and replaces it with a new edge from the root $r$ to $v$ in $G'$. The parameters of all vertices other than $v$ are unchanged. The parameters $\theta_v$ are set to the absolute pose (relative to $r$) of $v$ in $(G, \boldsymbol{\theta})$:

$$h_{\text{c}\to\text{f}}(G, v, r, \boldsymbol{\theta}) := (G', v, u', \boldsymbol{\theta}', f, f'),$$

where

$$(G', v, u') = g(G, v, r)$$
$$(f, f') = \text{proj}_{f, f'}(\theta_v)$$
$$\theta'_w = \theta_w \text{ for } w \neq v$$
$$\theta'_v = \mathbf{x}_v(G, \boldsymbol{\theta}) \quad \text{(pose of } v \text{ with respect to } r \text{ in } (G, \boldsymbol{\theta}))$$

**Floating to contact** This transition severs the edge from the parent of $v$ in $G$ (which is $r$) and replaces it with a new edge from another (object) vertex $u$ to $v$ in $G'$. The parameters of all vertices other than $v$ are unchanged. The parameters $\theta_v$ are computed by (i) computing the relative pose

$\Delta\mathbf{x}_{(u,f')\to(v,f)}(G,\boldsymbol{\theta}) \in SE(3)$ between the face $f$ of object $v$ (oriented according to its outward normal) and face $f'$ of object $u$ in $(G,\boldsymbol{\theta})$, and then (ii) transforming this pose into an element of $\mathbb{R} \times \mathbb{R} \times \mathbb{R} \times S^2 \times S^1$ via the function $\xi$ defined in Section O.4:

$$h_{\mathrm{f}\to\mathrm{c}}(G,v,u,\boldsymbol{\theta},f,f') := (G',v,r,\boldsymbol{\theta}'),$$

where

$$(G',v,r) = g(G,v,u)$$
$$\theta'_w = \theta_w \text{ for } w \neq v$$
$$\theta'_v = (f,f',a,b,z,\eta,\varphi)$$
$$(a,b,z,\eta,\varphi) = \xi(\Delta\mathbf{x}_{(u,f')\to(v,f)}(G,\boldsymbol{\theta}))$$

**Contact to contact** This transition severs the edge from the parent of $v$ in $G$ (which is some object $u'$) and replaces it with a new edge from another object $u$ to $v$ in $G'$. The parameters of all vertices other than $v$ are unchanged. The parameters $\theta_v$ are again computed by first computing the relative pose computing the relative pose $\Delta\mathbf{x}_{(u,f')\to(v,f)}(G,\boldsymbol{\theta}) \in SE(3)$, then applying $\xi$:

$$h_{\mathrm{c}\to\mathrm{c}}(G,v,u,\boldsymbol{\theta},f,f') := (G',v,u',\boldsymbol{\theta}',f_2,f'_2)$$

where

$$(G',v,u') = g(G,v,u)$$
$$\theta'_w = \theta_w \text{ for } w \neq v$$
$$\theta'_v = (f,f',a,b,z,\eta,\varphi)$$
$$(f_2,f'_2) = \mathrm{proj}_{f,f'}(\theta_v)$$
$$(a,b,z,\eta,\varphi) = \xi(\Delta\mathbf{x}_{(u,f')\to(v,f)}(G,\boldsymbol{\theta}))$$

**Proposition O.5.1.** *The function $h$ is an involution.*

*Proof.* First note that in each of the four cases ($z \in Z_i$ for $i = 1,2,3,4$), we have

$$\mathrm{proj}_{G,v,u}(h(z)) = g(\mathrm{proj}_{G,v,u}(z)).$$

Therefore, since $g$ is an involution, we have

$$\mathrm{proj}_{G,v,u}(h(h(z))) = g(\mathrm{proj}_{G,v,u}(h(z))) = g(g(\mathrm{proj}_{G,v,u}(z))) = \mathrm{proj}_{G,v,u}(z).$$

Next, note that if $z \in Z_2 \sqcup Z_4$, then $\mathrm{proj}_{f,f'}(h(h(z))) = \mathrm{proj}_{f,f'}(z)$ simply by unraveling the definitions.

It remains to show that $\mathrm{proj}_{\boldsymbol{\theta}}(h(h(z))) = \mathrm{proj}_{\boldsymbol{\theta}}(z)$. This clearly holds when $z \in Z_1$, as $h_{\mathrm{f}\to\mathrm{f}}$ is the identity.

For the case $z \in Z_2$, let $z = (G,v,r,\boldsymbol{\theta})$, and let $f,f' := \mathrm{proj}_{f,f'}(\theta_v)$, and let $G$ and $u$ be such that $g(G,v,r) = (G',v,u')$. Then, unraveling the definitions, we have

$$\mathrm{proj}_{\boldsymbol{\theta}}(h(z)) = \left\{ \begin{array}{ll} w \mapsto \theta_w & \text{for } w \neq v \\ v \mapsto \mathbf{x}_v(G,\theta) & \end{array} \right\}.$$

Unraveling the definitions one step further, we have

$$\mathrm{proj}_{\boldsymbol{\theta}}(h(h(z))) = \left\{ \begin{array}{l} w \mapsto \theta_w \quad \text{for } w \neq v \\ v \mapsto (f,f',\underbrace{\xi\left(\Delta\mathbf{x}_{(u',f')\to(v,f)}\left(G',\left\{\begin{array}{ll} w \mapsto \theta_w & \text{for } w \neq v \\ v \mapsto \mathbf{x}_v(G,\boldsymbol{\theta}) & \end{array}\right\}\right)\right)}_{:=\theta''_v := (a',b',z',\eta',\varphi')}) \end{array} \right\}.$$

So we need to show $\theta''_v = \theta_v$. By construction, $\theta''_v$ is the contact-parametrized relative pose for $v$ (face $f$) relative to $u'$ (the parent of $v$ in $G$, face $f'$) that, when converted to an absolute (relative to $r$) pose in the scene graph $(G,\boldsymbol{\theta})$, gives $\mathbf{x}_v(G,\theta)$. In other words, indeed $\theta''_v = \theta_v$.

For the case $z \in Z_3$, let $z = (G, v, u, \boldsymbol{\theta}, f, f')$ and $G' := \text{proj}_G(g(G, v, u))$. Unraveling two layers of definitions similarly to above, we have

$$\text{proj}_{\boldsymbol{\theta}}(h(h(z))) = \left\{ \begin{array}{l} w \mapsto \theta_w \quad \text{for } w \neq v \\ v \mapsto \underbrace{\mathbf{x}_v \left( G', \left\{ \begin{array}{l} w \mapsto \theta_w \quad \text{for } w \neq v \\ v \mapsto \Delta\mathbf{x}_{(u,f) \to (v,f')}(G, \boldsymbol{\theta}) \end{array} \right\} \right)}_{:= \theta_v''} \end{array} \right\}.$$

It again suffices to show $\theta_v'' = \theta_v$. By construction, $\theta_v''$ is the pose in world frame (relative to $r$) of object $v$ in $G'$; and the contact-parametrized relative pose of $v$ (face $f$) relative to $u$ (face $f'$) in $G'$ by construction has the property that, when converted to an absolute pose in $(G, \boldsymbol{\theta})$, the result is $\theta_v$. Thus $\theta_v'' = \theta_v$.

For the case $z \in Z_4$, let $z = (G, v, u, \boldsymbol{\theta}, f, f')$, and let $(f_2, f_2') := \text{proj}_{f,f'}(\theta_v)$, and let $G'$ and $u'$ be such that $(G', v, u') = g(G, v, u)$. Unraveling two layers of definitions again, we have

$$\text{proj}_{\boldsymbol{\theta}}(h(h(z))) = \left\{ \begin{array}{l} w \mapsto \theta_w \quad \text{for } w \neq v \\ v \mapsto \underbrace{(f, f', \xi \left( \Delta\mathbf{x}_{(u',f_2') \to (v,f_2)} \left( G', \left\{ \begin{array}{l} w \mapsto \theta_w \quad \text{for } w \neq v \\ v \mapsto (f, f', \xi \left( \Delta\mathbf{x}_{(u,f') \to (v,f)}(G, \boldsymbol{\theta}) \right), f, f') \end{array} \right\} \right) \right))}_{:= \theta_v'' := (a', b', z', \eta', \varphi')} \end{array} \right\}.$$

It again suffices to show $\theta_v'' = \theta_v$. Similarly to the above, $\theta_v''$ is a contact-parametrized relative pose for $v$ (face $f$) relative to $u'$ (the parent of $v$ in $G$, face $f'$) defined by the property that it produces the same absolute pose for $v$ as a second contact-parametrized relative pose. This second relative pose is for $v$ (face $f_2$) relative to $u'$ (face $f_2'$) that by construction produces the same absolute pose as $v$ has in $(G, \boldsymbol{\theta})$. It again follows that $\theta_v'' = \theta_v$, and this completes the proof. $\square$

The automated involutive MCMC implementation in Gen [6] includes an optional dynamic check that applies the involution twice to check that it is indeed an involution. We applied this check during testing of the algorithm to gain confidence in our implementation.

## O.6   The Radon–Nikodym derivative

The acceptance ratio for involutive MCMC [5] includes a "generalized Jacobian correction" term, equal to the Radon–Nikodym derivative of a pushforward measure $\mu_*$ with respect to a base measure $\mu$ defined on the state space $Z$ ($\mu$ is constructed the product measure of $\mu_P$ and $\mu_Q$ [5]). Next, $\mu_* := \mu \circ h^{-1}$ is the pushforward of $\mu$ by the involution $h : Z \to Z$ described above. To justify the validity of this involutive MCMC kernel, we must show that $\mu_*$ is absolutely continuous with respect to $\mu$, i.e., that the Radon–Nikodym derivative $\frac{d\mu_*}{d\mu}$ exists. Because all the discrete choices in the model (graph structure, contact faces, etc.) are assigned positive probability mass in both the model and the proposal, it suffices to show absolute continuity for the continuous part of the involution: the mapping (call it $\ell^\circ$) that, for given contact faces $f, f' \in F$, converts between a 6DoF pose $\theta_1 \in SE(3)$ and a contact-parameterized relative pose $\theta_2 = (a, b, z, \eta, \varphi) \in \mathbb{R} \times \mathbb{R} \times \mathbb{R} \times S^2 \times S^1$ (note that in this section we use $\theta_2$ to denote only the continuous part of the contact-parameterized relative pose).

Note that $\ell^\circ$ depends on not just $\theta_1$, but also on the scene graph, objects and faces $(G, \boldsymbol{\theta}, v, u', f, f')$. Specifically, the absolute pose $\theta_1$ is gotten by pre- and post-composing $\Delta\mathbf{x}_{(u',f') \to (v,f)}(G, \boldsymbol{\theta})$ with rigid motions that depend on the absolute poses of face $f'$ of $u$ and face $f$ of $v$ in scene graph $(G, \boldsymbol{\theta})$, but these rigid motions do not depend on $\theta_1$ or $\theta_2$ themselves. Thus, in the sections below, rather than $\ell^\circ$ itself, we analyze $\ell$, the variant of $\ell^\circ$ which operates on 6DoF relative poses $\Delta\mathbf{x}$ where $\ell^\circ$ operates on 6DoF absolute poses $\mathbf{x}$. Because rigid transformations are diffeomorphisms and their Radon–Nikodym derivatives (Jacobian determinants) are identically 1, the results in the sections below, which show that $\ell$ has a Radon–Nikodym derivative that is piecewise constant on $A \sqcup B$ (defined below), apply equally well to the map $\ell^\circ$ which parameterizes $\theta_1$ relative to the world coordinate frame in some particular scene graph.

We can denote a rigid motion by the pair $(\mathbf{t}, \omega)$, where $\mathbf{t} \in \mathbb{R}^3$ is the translation component and $\omega \in SO(3)$ is the rotation component. (In algebraic terms, we are identifying $SE_3$ with the semidirect

product $\mathbb{R}^3 \rtimes SO(3)$.) Accordingly, define projection functions $\mathrm{proj}_{\mathbf{t}}$ and $\mathrm{proj}_\omega$ on $SE(3)$ as in Section O.1.

Let $Z := A \sqcup B$ where $A := \mathbb{R}^3 \times SO(3)$ and $B := \mathbb{R} \times \mathbb{R} \times \mathbb{R} \times S^2 \times S^1$, and let $\nu$ denote the base measure on $Z$.[3] In the sections below, we discuss the pushforward $\nu_* := \nu \circ \ell^{-1}$ and its Radon–Nikodym derivative $\frac{d\nu_*}{d\nu}$.

### O.6.1   Existence of the Radon–Nikodym derivative

In this section, we prove $\nu_* \ll \nu$. Because $\nu_* = \nu \circ \ell^{-1}$, the proof proceeds by analyzing the involution $\ell$. First we show that $\ell$ is defined almost everywhere, so that $\nu_*$ is well-defined. Then we show that there exists a subset $Z'' \subseteq Z$ whose complement has measure zero, such that the restriction of $\ell$ to $Z''$ is a diffeomorphism. It follows that $\nu_* \ll \nu$ by [9, Prop 6.5], since $\ell^{-1}$ is a smooth map.

First, to show that $\nu_*$ is well-defined, we show that $\ell$ is defined almost everywhere on $Z$. Indeed, the domain of $\ell$ is $A' \sqcup B'$, where $A' := \mathbb{R}^3 \times \mathrm{domain}(\xi)$ and $B' := \mathbb{R} \times \mathbb{R} \times \mathbb{R} \times \mathrm{domain}(\xi^{-1})$ (where $\xi$ is as defined in Section O.4). Now, $\mathrm{domain}(\xi) = \{\omega \in SO(3) : \omega(0,0,1)^\top \neq (0,0,-1)^\top\}$ has a complement of measure zero in $SO(3)$, and $\mathrm{domain}(\xi^{-1}) = (S^2 \setminus \{(0,0,-1)^\top\}) \times S^1$ has a complement of measure zero in $S^2 \times S^1$, so indeed $\mathrm{domain}(\ell) = A' \sqcup B'$ has a complement of measure zero in $Z$.

Next, we show that there exist subsets $A'' \subseteq A'$, $B'' \subseteq B'$, whose complements also have measure zero, such that $\ell$ fixes $A \sqcup B$ setwise and the restriction of $\ell$ to $A'' \sqcup B''$ is a diffeomorphism. First, note that the coordinates $\mathbf{t}$ in $A$ and the coordinates $a, b, z$ in $B$ represent the same translation in a different coordinate frame. Thus, for any fixed $\omega \in SO(3)$, the function $\mathbf{t} \mapsto \mathrm{proj}_{a,b,z}(\ell(\mathbf{t}, \omega))$ is a rigid motion, hence a diffeomorphism. Furthermore, the rotation component of $\ell(\theta)$ (regardless of whether $\theta \in A$ or $\theta \in B$) depends only on the rotation component of $\theta$, not at all on the translation component. Thus, $\ell$ is a diffeomorphism from $A''$ to $B''$ if and only if $\ell^\star$ is a diffeomorphism from $\mathrm{proj}_\omega(A'')$ to $\mathrm{proj}_{\eta,\varphi}(B'')$, where $\ell^\star(\omega) := \mathrm{proj}_{\eta,\varphi}(\ell(0,\omega))$. Taking $A'' := \mathbb{R}^3 \times A^\star$ and $B'' := \mathbb{R} \times \mathbb{R} \times \mathbb{R} \times B^\star$, we see that it suffices to find subsets $A^\star \subseteq SO(3)$ and $B^\star \subseteq S^2 \times S^1$ whose complements have measure zero, such that the restriction of $\ell^\star$ to $A^\star$ is a diffeomorphism onto $B^\star$.

Denote elements of $SO(3)$ as $\omega = \mathrm{spin}(w, x, y, z)$, where $\mathrm{spin} : S^3 \to SO(3)$ is the 2-to-1 smooth covering map that carries a unit quaternion $w + x\mathbf{i} + y\mathbf{j} + z\mathbf{k}$ to its corresponding rotation. Then, we take

$$A^\star = \big\{\mathrm{spin}(w,x,y,z) \,\big|\, (w,x,y,z) \in S^3;\ z \neq 0\big\}$$
$$B^\star = \big\{((a,b,c),\varphi) \in S^2 \times S^1 \,\big|\, c \neq \pm 1\big\}$$

To show that $\ell$ is a diffeomorphism from $A^\star$ to $B^\star$, we give an explicit formula for $\ell^\star$ in terms of coordinates below (Section O.6.2).

### O.6.2   Formula for the mapping in coordinates

In this section we give an explicit formula for the map $\ell^\star$ defined in SectionO.6.1 in terms of coordinates. For elements $\omega = \mathrm{spin}(w,x,y,z) \in A^\star$, the $S^2$ component of $\ell^\star(\omega)$ is the image of $(0,0,1)^\top$ under the rotation, and is given by [8, §8.2]:

$$\eta = \mathrm{spin}(w,x,y,z) \begin{pmatrix} 0 \\ 0 \\ 1 \end{pmatrix} = \begin{pmatrix} 2(xz + wy) \\ 2(yz - wx) \\ 1 - 2(x^2 + y^2) \end{pmatrix}.$$

Even though $(w,x,y,z)$ is not uniquely determined by $\mathrm{spin}(w,x,y,z)$, the above expression is well-defined because both possible choices of quaternion—$(w,x,y,z)$ and $(-w,-x,-y,-z)$—give the same value for the right-hand side.

To compute the $S^1$ component $\varphi$, note that the set of all rotations that carry $(0,0,1)^\top$ to $\eta$ is

$$\{\mathrm{spin}(w,x,y,z) \circ R_{(0,0,1)}(-\varphi') : 0 \leq \varphi' < 2\pi\},$$

---

[3]That is, the sum of (i) the product of Lebesgue measure on $\mathbb{R}^3$ and Haar measure on $SO(3)$, and (ii) the product of Lebesgue measure on $\mathbb{R} \times \mathbb{R} \times \mathbb{R}$, spherical uniform measure on $S^2$, and uniform spherical measure on $S^1$.

where $R_{(0,0,1)}(\varphi')$ is a rotation about the axis $(0,0,1)^\top$ by angle $\varphi'$. Because the action of $S^3$ on itself by quaternion multiplication is a geometric rotation of $S^3$ (in particular, an isometry) [8, §8.3], minimizing geodesic distance among the above family of rotations is equivalent to minimizing (over $\varphi'$) geodesic distance from $R_{(0,0,1)}(\varphi')$ to $(w,x,y,z)$. Explicitly, $R_{(0,0,1)}(\varphi')$ corresponds to the unit quaternions

$$\pm \left( \cos(\varphi'/2),\ 0,\ 0,\ \sin(\varphi'/2) \right).$$

Note that minimizing geodesic distance on the sphere $S^3$ is equivalent to minimizing the cosine between the corresponding vectors in $S^3 \subseteq \mathbb{R}^4$. By the cosine double angle formula, the cosine between $R_{(0,0,1)}(\varphi')$ and $(w,x,y,z)$ in this sense is

$$2 \left( \pm \left( \cos(\varphi'/2),\ 0,\ 0,\ \sin(\varphi'/2) \right) \cdot (w,x,y,z) \right)^2 - 1,$$

where $\cdot$ denotes the dot product in $\mathbb{R}^4$. This quantity is maximized precisely when the doct product is either maximized or minimized, so we can drop the $\pm$. We can then compute the minima and maxima of the dot product by setting the derivative equal to zero: we have

$$\left( \cos(\varphi'/2),\ 0,\ 0,\ \sin(\varphi'/2) \right) \cdot (w,x,y,z) = w\cos(\varphi'/2) + z\sin(\varphi'/2)$$

and the above expression is minimized or maximized when

$$\varphi'/2 = \arctan(z/w) + \pi n \quad \text{for some } n \in \mathbb{Z}$$

or equivalently, $\varphi' = 2\arctan(z/w) + 2\pi n$. Thus, the $S^1$ component of $\ell^\star(\mathrm{spin}(w,x,y,z))$ that we set out to compute is

$$\varphi = 2\arctan(z/w),$$

where the branch cut in $\arctan$ is chosen so that the output lies in the interval $[0,\pi)$, and we allow $z/w$ to lie on the extended real line, with $\arctan(\pm\infty) = \pi/2$.

Since $z \neq 0$ in $A^\star$, $\varphi$ never lands on the branch cut. Thus, $\ell^\star(\mathrm{spin}(w,x,y,z))$ is a smooth function of the quaternion $(w,x,y,z)$. Because spin is a smooth covering map [9, ch. 4], it follows[4] that spin is also a smooth function of the element $\omega = \mathrm{spin}(w,x,y,z) \in SO(3)$.

The above definition expresses in coordinates the geometry of Hopf fibration and choice of branch cut described in Section O.4. We now need only show that $\ell^\star$ has a smooth inverse. For elements $(\eta, \varphi) \in B^\star$, we take [2, §7]

$$(\ell^\star)^{-1}((a,b,c),\varphi) =$$
$$\mathrm{spin}\left( \tfrac{1}{\sqrt{2(1+c)}} \left( (1+c)\cos(\varphi),\ a\sin(\varphi) - b\cos(\varphi),\ a\cos(\varphi) + b\sin(\varphi),\ (1+c)\sin(\varphi) \right) \right).$$

This function is clearly smooth on $B^\star$, and direct computation shows that $(\ell^\star)^{-1} \circ \ell^\star$ is the identity.

### O.6.3  Value of the Radon–Nikodym derivative

In the preceding sections we showed that $\nu_* := \nu \circ \ell^{-1}$ has a density with respect to $\nu$, the Radon–Nikodym derivative $\rho := \frac{d\nu_*}{d\nu}$. In this section we argue that $\rho$ is (a.e.) constant on each of the connected components $A$ and $B$. Because $h$ acts as isometries on the translation components, and acts on rotation components in a way that doesn't depend on the translation components, we need only look at orientation components. That is, the Radon–Nikodym derivative $\rho$ is equal to the Radon–Nikodym derivative of the pushforward $\nu_*^\star$ of the base measure[5] $\nu^\star$ on $A^\star \sqcup B^\star$ by $\ell^\star$:

$$\frac{d\nu_*}{d\nu}(\mathbf{t},\omega) = \frac{d\nu_*^\star}{d\nu^\star}(\omega) \quad \text{and} \quad \frac{d\nu_*}{d\nu}(a,b,z,\eta,\varphi) = \frac{d\nu_*^\star}{d\nu^\star}(\eta,\varphi).$$

Note the following chain of equivalences: $\frac{d\nu_*^\star}{d\nu^\star}$ is a.e. constant on $A^\star \iff \nu_*^\star$ is a scalar multiple of the Haar measure on $A^\star \iff \nu_*^\star$ is invariant under the action of $SO(3)$ on itself by multiplication.

---

[4]This can be seen by pre-composing $h''$ with a lifting to one of the sheets in an evenly covered neighborhood of $\omega$.

[5]In this case, sum of the Haar measure on $A^\star \subseteq SO(3)$ and the product of uniform measures on $B^\star \subseteq S^2 \times S^1$.

We can see that the latter statement holds by noting three things: First, the Haar measure on $SO(3)$ equals the pushforward by spin of the Haar measure on unit quaternions. Next, the action of $S^3$ on itself by group multiplication is an action by isometries [8, §8.3]. Finally, by [14], the volume element on $S^3$ equals the product of the volume elements on $S^2$ and $S^1$ (here we are using the fact that the Haar measure on $S^3$ coincides with the Borel measure when it is viewed as a Riemannian manifold). Thus the action of $SO(3)$ on itself by multiplication, when pushed through $(\ell^\star)^{-1}$, becomes an action by local isometries on $S^2 \times S^1$, and is thus invariant under the base measure $\nu^\star$ (which on $B^\star$ is the product of uniform measures). Therefore indeed $\frac{d\nu_*^\star}{d\nu^\star}$ is a.e. constant on $A^\star$. Since $\ell^\star$ is an involution and $\ell^\star(A^\star) = B^\star$, it follows that $\frac{d\nu_*^\star}{d\nu^\star}$ is a.e. constant on $B^\star$, and the values of the Radon–Nikodym derivative on $A^\star$ and $B^\star$ are reciprocals of each other.

### O.6.4 Acceptance probability

For our choice of scaling constants, if we assign total measures $SO(3) \mapsto \pi^2$, $S^2 \mapsto 4\pi$, and $S^1 \mapsto 2\pi$, then the Radon–Nikodym derivative corrections in the involutive MCMC acceptance probability [5] are: 1 (for 'floating to floating' and 'contact to contact' moves), $(4\pi \cdot 2\pi)/\pi^2 = 8$ (for a 'floating to contact' move), and $\pi^2/(4\pi \cdot 2\pi) = 1/8$ (for a 'contact to floating' move). This gives the following acceptance probabilities for each of the four possible proposed moves from $(G, \boldsymbol{\theta})$ to $(G', \boldsymbol{\theta}')$ that can be proposed within our kernel:

**Floating to floating** When $u = r$ and $u' = r$, the state is unchanged, and the move always accepts:

$$\alpha = \min\left\{1, \frac{p(\mathbf{Y}|N, \mathbf{c}, G, \boldsymbol{\theta})}{p(\mathbf{Y}|N, \mathbf{c}, G, \boldsymbol{\theta})}\right\} = 1 \tag{23}$$

**Floating to contact** When $u \neq r$ and $u' = r$ (we are severing $v$ from the root and grafting $v$ onto another object), the acceptance probability is:

$$\alpha = \min\left\{1, \frac{p(\mathbf{Y}|N, \mathbf{c}, G', \boldsymbol{\theta}')}{p(\mathbf{Y}|N, \mathbf{c}, G, \boldsymbol{\theta})} \frac{p(\boldsymbol{\theta}'|N, \mathbf{c}, G')}{p(\boldsymbol{\theta}|N, \mathbf{c}, G)} \frac{q(v', u'; G')}{q(v, u, f_1, f_1'; G)} \cdot 8\right\} \tag{24}$$

where $(f_1, f_1') = \mathrm{proj}_{f,f'}(\theta_v')$.

**Contact to floating** When $u = r$ and $u' \neq r$ (we are severing $v$ from an object and grafting $v$ onto the root), the acceptance probability is:

$$\alpha = \min\left\{1, \frac{p(\mathbf{Y}|N, \mathbf{c}, G', \boldsymbol{\theta}')}{p(\mathbf{Y}|N, \mathbf{c}, G, \boldsymbol{\theta})} \frac{p(\boldsymbol{\theta}'|N, \mathbf{c}, G')}{p(\boldsymbol{\theta}|N, \mathbf{c}, G)} \frac{q(v', u', f_2, f_2'; G')}{q(v, u; G)} \cdot \frac{1}{8}\right\} \tag{25}$$

where $(f_2, f_2') = \mathrm{proj}_{f,f'}(\theta_v)$.

**Contact to contact** When $u \neq r$ and $u' \neq r$ (we are severing $v$ from an object and grafting $v$ onto another object), the acceptance probability is:

$$\alpha = \min\left\{1, \frac{p(\mathbf{Y}|N, \mathbf{c}, G', \boldsymbol{\theta}')}{p(\mathbf{Y}|N, \mathbf{c}, G, \boldsymbol{\theta})} \frac{p(\boldsymbol{\theta}'|N, \mathbf{c}, G')}{p(\boldsymbol{\theta}|N, \mathbf{c}, G)} \frac{q(v', u', f_2, f_2'; G')}{q(v, u, f_1, f_1'; G)}\right\} \tag{26}$$

where $(f_1, f_1') = \mathrm{proj}_{f,f'}(\theta_v')$ and $(f_2, f_2') = \mathrm{proj}_{f,f'}(\theta_v)$.

### O.7 Lemmas

**Lemma O.7.1.** *Let $G = (V, E)$ be a directed tree rooted at $r \in V$, and suppose $u, u', v \in V$ are such that the following conditions hold:*

    *(i) $u \neq v$*
    *(ii) $u$ is not a descendant of $v$*
    *(iii) $(u', v) \in E$.*

*Let $G'$ be the directed graph obtained from $G$ by deleting the edge $(u', v)$ and adding the edge $(u, v)$, that is, $G' = (V, (E \setminus \{(u', v)\}) \cup \{(u, v)\})$. Then $G'$ is a directed tree rooted at $r$.*

*Proof.* We show that for any vertex $w$, there is a unique path in $G'$ from $r$ to $w$.

First, suppose $w$ is not a descendant of $v$ in $G'$. Then $w$ is not a descendant of $v$ in $G$, since the set of descendants of $v$ is the same in $G$ and $G'$. Thus no path from $r$ to $w$ in either $G$ or $G'$ passes through $v$; consequently, no path from $r$ to $w$ in either $G$ or $G'$ contains either of the edges $(u'v)$ or $(u, v)$. Thus, a sequence of vertices $r = x_0$, $x_1$, ..., $x_n = w$ is a path in $G'$ if and only if it is a path in $G$. Since there is a unique path from $r$ to $w$ in $G$, it follows that there is a unique path from $r$ to $w$ in $G'$.

Next, suppose $w$ is a descendant of $v$ in $G'$ (and hence also in $G$). Because $u'$ is the only in-neighbor of $v$ in $G$, it follows that $u$ is the only in-neighbor of $v$ in $G'$. Thus, every path from $r$ to $w$ in $G'$ is the concatenation of a path from $r$ to $u$ with a path from $v$ to $w$. But since $u$ is not a descendant of $v$ in $G$ (hence neither in $G'$), there is a unique path from $r$ to $u$ in $G'$, by the above paragraph. And of course, there is a unique path from $v$ to $w$ in $G$, hence also in $G'$ since the subtrees rooted at $v$ are the same. Therefore there is a unique path from $v$ to $w$ in $G'$. $\square$