# OpenReview forum: "3DP3: 3D Scene Perception via Probabilistic Programming"
_NeurIPS.cc/2021/Conference — NeurIPS 2021 Poster_

### Official Review · Reviewer_NB9L · 2021-07-15

**Rating:** 7
**Confidence:** 3

**Summary:**

This paper proposes a probabilistic model for reconstructing a scene of multiple 3D objects. The key advantage of the approach is that it is able to reason about the shapes of objects, the relations between them (via the scene graph) and their contacts. This ability to use physical constraints for scene parsing is a major limitation of deep learning systems. The results show that the proposed method is significantly more accurate than two recent approaches that are based on deep models (i.e. DenseFusion and Robust6D).  The main limitation of the model seems to be the inference speed, which is currently not realtime.


**Limitations And Societal Impact:**

The paper could do with a better description of the limitations of the approach (as mentioned above), and there is no mention of the potential negative societal impacts.

**Main Review:**

**TL;DR:** An intriguing, principled, probabilistic approach for object shape and pose estimation that is able to utilize object relations and physical constraints. In the main review below I have labelled each point as positive (+), negative (-) or mixed (+/-).

**Originality**:

(+) As far as I can tell the method is new in the sense that this is the first probabilistic, generative model that takes into account both object shapes, physical constraints such as their spatial relations and contact between individual objects. This is interesting as physical reasoning can help to overcome the limitations of traditional object reconstruction methods that only take into account multi-view geometry and simple shape priors. The results presented in the paper confirm this as the proposed system is significantly more accurate than the two approaches against which it is compared.

(+) Although this type of analysis-by-synthesis approach based on MCMC has been very popular and well explored in computer vision in the past, the probabilistic model presented in the paper is novel. Furthermore, the meticulously crafted MCMC kernels used for inference are also novel and seem to be effective.

(+/-) In terms of related work, the authors do a good job at describing which contributions are novel. I think the related work Section 2 on “Probabilistic programs for vision” is particularly clear and gives a concise description of what is novel compared to general probabilistic programming works that have addressed computer vision problems. However, I would probably like to have seen a discussion about more closely related work such as [A,B] that are probabilistic approaches for addressing the same problem described in this paper (i.e. joint scene parsing and object reconstruction). At the moment the related work doesn’t cover such approaches.

[A] Du, Yilun, et al. "Learning to Exploit Stability for 3D Scene Parsing." NeurIPS. 2018.

[B] Bao, Sid Yingze, and Silvio Savarese. "Semantic structure from motion: A novel framework for joint object recognition and 3D reconstruction." Outdoor and Large-Scale Real-World Scene Analysis. Springer, Berlin, Heidelberg, 2012. 376-397.

**Quality**:

(+) Although I have not checked every detail of the proofs in the submission (particularly the ones in the supplementary), the general approach appears to be technically sound as far as I can tell.

(+) The claims in the paper are well supported. Particularly the results show that the proposed method is able to learn new objects from a very small number of views i.e. 5 and using this model is able to obtain state of the art pose estimation results on the YCB dataset.

(-) There is a caveat with these results though, in that the method appears to be initialized with a state-of-the-art deep learning model, which makes the results slightly less impressive.  However this still shows that the proposed approach is able to refine the poses to a level beyond that which can be obtained with current methods. I would suggest that the authors make it clear in the paper where the method has been initialized using an existing approach and where it hasn’t because this isn’t very clear at the moment. Also, what is the performance of the system when no initialization is used?

(-) Overall I would say that the authors could do a better job of describing the weaknesses of the approach. The most obvious weakness that comes to mind is computational efficiency. There should probably be some form of inference time comparison in the paper to show how the proposed approach compares to the existing methods (especially deep learning based ones) which are known to be very fast. Another weakness is the complexity of the approach - is it not possible to simplify the components such as the transition kernels or the scene graph structure and obtain similar results?

**Clarity**:

(-) The paper is very dense and not always that easy to follow.  This is partly due to the nature of the approach - the model contains many quantities and distributions that need to be defined, but it might be useful for the reader if there was some summary or overview that describes all the components of the method in one place.

(+/-) Overall, the paper is well organized and the individual sections are quite verbose and not that easy to follow. I don’t have a great suggestion to fix this, but maybe separating the definitions from the actual description might help.

(+) As far as I can tell, the paper provides enough information for an expert to reimplement the method (with the help of a probabilistic programming language, like Gen) and reproduce results. The authors have also supplied the source code which helps with reproducibility.

**Significance**:

(+) In my view, the results presented in the paper are significant. This is one of the best examples of how probabilistic programming can be applied to solve challenging vision tasks.  It is quite surprising how data-efficient and accurate the approach is compared to deep learning systems that need many training samples to achieve similar accuracy for object pose estimation.
I think the great results obtained in the paper could help to encourage more computer vision researchers to explore probabilistic programming approaches. There are also quite a few areas in which the proposed method could be improved (such as inference time etc.) which might inspire other researchers to try to improve upon the method.

(+) In summary, the method addresses a difficult task of jointly estimating object shape and pose, and also  advances the state of the art in terms of accuracy.

# Post author response and discussion
After considering the other reviews, and the authors' responses, the main concerns raised in my review have been addressed. Specifically, the issues related to how the initialization is done and a more thorough discussion about the weaknesses of the approach have been addressed. In terms of accessibility / reproducibility the authors have committed to open-sourcing their approach. Overall, I think this is a good paper which opens up lots of possibilities for future research at the intersection of probabilistic modelling and computer vision. Therefore, my final rating is "7: Good paper, accept".


**Time Spent Reviewing:**

2

---

> ### Author Response · Authors · 2021-08-10
> **Author Response**
>
> Thank you for your thorough review, and for your encouraging comments on the novelty of our approach in accounting for both object shape and physical constraints and the significance of our improvements in data-efficiency and accuracy.
>
> **I would probably like to have seen a discussion about more closely related work such as [A,B] that are probabilistic approaches for addressing the same problem described in this paper (i.e. joint scene parsing and object reconstruction).**
>
> Thank you for pointing us to these two important related works. We have added a discussion of  [A] and [B] to our Related Work section:
>
> [A] Du et al.
>
> Summary: our system is more data efficient and encodes a complementary inductive bias (based on geometric contact but not physical stability).
> Du et al. focused on improving accuracy and reducing the amount labeled training data required for pose estimation and scene parsing by using physical stability as part of the training objective for a neural network. Compared to that work, our system requires vastly less training data (5 views versus tens of thousands of training examples) and is based on a probabilistic generative model rather than a neural network. Both our work and Du et al. utilize prior knowledge about object interactions to improve accuracy of scene parsing. It would be interesting to draw on the approach of Du et al. and incorporate a soft constraint based on a stability test using physical simulation into our probabilistic generative model to encourage scenes that are physically stable. (Our use of scene graph structure prior tends to place more probability mass on physically stable configurations than a prior that samples each object's 6DoF pose independently, but it is not designed specifically to generate physically stable scene parses).
>
> [B] Bao et al.
>
> Bao et al. combine object detectors (pre-trained on a large corpus) with a handful of views of each test scene to produce scene parses that are made more robust by exploiting the geometric constraints of consistency across viewpoints.
> By contrast, our work combines object models learned from 5 training images per object, with a single view of each test scene, to produce parses that are made robust by a different kind of geometric constraint.
>
> **(-) There is a caveat with these results though, in that the method appears to be initialized with a state-of-the-art deep learning model, which makes the results slightly less impressive. ... Also, what is the performance of the system when no initialization is used?**
>
> All the evaluations on the YCB-Challenging dataset (Figure 6, Table 1, Supplementary Table 3) did not use initialization from deep learning. Instead we use our own initialization procedure described in Supplement Section F. Only for the evaluation on YCB-Video did we use the deep learning model to initialize our inference. We will clarify this in the final paper.
>
> The YCB-Video dataset contains scenes with many objects close together and noisy depth data. As a result, the initialization procedure in Appendix F struggles to disambiguate object instances since it only uses the depth input and ignores the RGB input. DenseFusion leverages the RGB input to detect object instances and give initializations of their poses that our system can then refine. We have adapted our scene graph inference algorithm to incorporate RGB data into the likelihood model and early results indicate that incorporating RGB likelihoods makes our method less sensitive to having an accurate initialization (see [link](https://bit.ly/3m45VGb) for some qualitative results from our adapted algorithm that were obtained without DenseFusion initialization). However, we do believe that deep neural inference models like DenseFusion are in general complementary to our scene graph inference algorithm and are an important component in real-time perception.
>
> **Overall I would say that the authors could do a better job of describing the weaknesses of the approach. The most obvious weakness that comes to mind is computational efficiency. There should probably be some form of inference time comparison...**
>
> Thank you for these suggestions. We agree that running time is a potential limitation of the approach and we will add a deeper discussion of this in the paper.
>
> In response to your suggestion, we are running a quantitative evaluation of inference time of our method and all the baselines on the YCB datasets and will include these results in the paper. Some of the preliminary results here show the following times for each method: DenseFusion: ~0.5s / frame, 3DP3 (scenes with 2 objects):  ~5s / frame, 3DP3 (scenes with 4 objects): ~10s / frame. While our method is slower than DenseFusion, it is substantially more accurate, and our inference times are comparable to other approaches that use depth rendering in the loop (including GRIP [1] which reports ~10s per frame). While these frame rates are clearly insufficient for some applications like autonomous vehicles, many robotics tasks do not require high frame rates (see e.g. the influential paper [2], which also uses ~10 seconds to perform perception before planning a grasp by a robot manipulator).
>
> That said, we believe that our method can be optimized for use in some real-time applications, via running the likelihood computation on a GPU and via coarse-to-fine inference. Note that our running time is dominated by the likelihood computation. In the modified version of our method  that we described above in the discussion of Eslami et al, the RGB-based likelihood is more efficient and gives running times in the range of ~0.5s / frame ([link](https://bit.ly/3m45VGb)).
>
> [1] Chen, Xiaotong, et al. IROS 2019
>
> [2] Maitin-Shepard et al., ICRA 2020
>
> **Another weakness is the complexity of the approach - is it not possible to simplify the components such as the transition kernels or the scene graph structure and obtain similar results?**
>
> We recognize that our scene graph structure is complex relative to the latent-variable representations of other generative models for scene understanding (e.g. Eslami et al. NeurIPS 2016). However, we believe that the need for this structure to explain complex scenes parsimoniously is well-motivated by the widespread use of hierarchical scene graph structures to manage complexity of scene representation in game engines and computer graphics going back to [3]. We also note that the complexity, once incurred, pays dividends via its productivity -- very complex scenes can be described simply as more complex instances of scene graphs.
>
> Much of the complexity of the transition kernel is involved in changing parametrization of 6DoF poses. These change-of-variable transformations need only be implemented once within a library. Indeed, we have built a scene graph library for the Gen probabilistic programming system that provides all of the building blocks used in this paper as reusable modules.
>
> Future work will explore tradeoffs between simple heuristics for scene graph inference; neural scene graph proposals; and the MCMC approaches we pursue here. The sparser the data, and the more safety-critical the application, the more room there is for the complexity of Bayesian scene graph inference to yield payoffs in uncertainty quantification over a broad range of problem instances.
>
> [3] James H. Clark. "Hierarchical geometric models for visible surface algorithms. Communications of the ACM Volume 19 Issue 10 Oct. 1976.
>
> **The paper is very dense and not always that easy to follow. ... Overall, the paper is well organized and the individual sections are quite verbose and not that easy to follow. I don’t have a great suggestion to fix this, but maybe separating the definitions from the actual description might help.**
>
> Thank you for these comments. We will add a table that summarizes the key notations and where in the text they are introduced to help readers navigate the paper.
>
> **The paper could do with a better description of the limitations of the approach (as mentioned above), and there is no mention of the potential negative societal impacts.**
>
> Thank you for these suggestions. We will add a deeper discussion of the limitations as described above.
>
> We have added the following Societal Impact section at the end of the paper:
>
> 3D scene graph perception is potentially useful as a central computer vision component in diverse application domains such as AR, security/surveillance, HCI, and robotics. This paper shows how to improve robustness, accuracy, and learnability/data efficiency, relative to pure deep learning approaches. These improvements could make it easier to build and maintain beneficial systems as well as harmful ones. Examples of potentially beneficial applications include assistive indoor AR/mixed-reality for the cognitively impaired; some indoor safety/security applications; and educational technology. Examples of potentially harmful applications include widespread, automated analysis of humans and their interactions with the physical environment, by nation states and for-profit corporations, and the development of improved indoor perception & target tracking systems for autonomous weapons.
>
> This paper also opens up new avenues for research, as well as new potential risks, in dataset bias and disparate impact for 3D perception, via its use of human-editable probabilistic programs as scene graph priors.

---

> > ### Comment · Reviewer_NB9L · 2021-09-02
> > **Re: author response**
> >
> > Thanks for the detailed response. The main concerns raised in my review have been addressed. Particularly, the points related to what initialization is used when have been clarified and some discussion about the weaknesses of the approach (including the inference times) have been added.
> >
> > Overall, I think this is a really good paper. However, one general, remaining critique I have is that the complexity of the approach might make it inaccessible to a large part of the machine-learning / robotics community - but this is more a comment on the audience it could reach than a problem with the paper itself.

---

> > > ### Author Response · Authors · 2021-09-04
> > > **Response**
> > >
> > > Thank you, we appreciate your response!
> > >
> > > We are deeply invested in ensuring our work is accessible to the broader machine-learning / robotics community. First, we commit to open sourcing 3DP3 and providing an easy-to-use implementation and interface that will enable researchers to use 3DP3 within their vision pipelines. In addition, we will provide all the separate modules (e.g. rendering-based likelihoods, pose distributions, scene graph representation) that will enable researchers to build upon the 3DP3 framework we have developed and modify it for their specific application.
> > >
> > > For the paper itself, in the camera-ready version we plan to use part of the additional content page to provide a summary/overview of the model and each of its components. We will also make a clearer separation of definitions from descriptions as we introduce each model component.

---

### Official Review · Reviewer_cyy4 · 2021-07-17

**Rating:** 6
**Confidence:** 3

**Summary:**

The paper presents an approach for inferring the 3D pose and shape of multi-object tabletop scenes from a set of multi-view depth frames as input.  The approach is based on a generative model incorporating 3D scene graph generation, as well as per-object category voxelized shape priors.  At test time, the inference algorithm parses the input to a 3D scene graph using MCMC-based updates to the scene graph structure, constrained by physical contact relations between the objects and the object shape priors.

The approach is evaluated on the 6DoF 3D pose estimation task using two datasets: a set of real scenes from the YCB-video dataset, and a set of synthetically generated scenes using reconstructed YCB objects (YCB-Challenging).  The experiments compare the proposed approach against baselines from prior work (specifically, DenseFusion and Robust6D -- both trained in a supervised fashion on larger RGB-D training datasets).  An ablation of the approach without scene graph structure inference is also used as a comparison point.  The experiments show that the approach mostly outperforms the baselines and ablation in terms of accuracy of the predicted 6DoF pose for the objects in the scene.  An additional analysis incorporates one of the baselines (DenseFusion) to initialize the estimated pose used by the proposed method. Improvements to the estimated pose are observed in some challenging scenarios from the YCB-Video test scenes.

**Limitations And Societal Impact:**

Some limitations are discussed at a high level in the last section.  The speed of inference, and integrating object prior learning with scene graph parsing were identified as avenues for future work.  Potential negative societal impacts were not discussed (the checklist notes that the authors do not believe there are any at this stage of the work).

**Main Review:**

Originality:
On the plus side, the proposed approach is quite comprehensive and formalized in a way that admits an impressive level of generality.  As such, I can see the approach being applicable to domains and datasets beyond what has been presented in the paper.  I believe that the community would benefit from more work in this vein, leveraging probabilistic programming and generative models for 3D scene understanding.  On the other hand, there is a highly relevant line of prior work that is not cited or discussed: "Attend, Infer, Repeat:
Fast Scene Understanding with Generative Models" by Eslami et al. NIPS 2016 have proposed a similar probabilistic inference framework and demonstrated that it can be used to parse 3D scenes (i.e. infer the object that are present, and their 6DoF poses).  A detailed discussion of how the proposed approach relates to this prior work, and potentially comparison against it in the evaluation is in my opinion necessary.

Quality:
The paper is technically sound, as far as I can tell and the claims are mostly supported by the empirical evidence.  That being said, there is no evaluation of the object shape priors either independently, or as part of the proposed approach.  In other words, the impact of Section 4 on the results is not clear in the current paper.

Clarity:
Overall, the paper is very well written and organized coherently.  There were a few minor typos etc. that could be resolved in a proofreading pass.  A couple of examples that I detected: L98 "corresponds to rigid object with" should be "corresponds to a rigid object with"; and L201 "a uninformed" should be "an uninformed".

Significance:
The paper has an impressive scope in terms of the presented formalism and its generality.  However, the empirical results only partially back up the complexity of the proposed approach, and some discussion of relation to prior work is missing (see above points).  Therefore, I am not as positive as I could be with respect to acceptance.

**Time Spent Reviewing:**

2

---

> ### Author Response · Authors · 2021-08-10
> **Author Response**
>
> Thank you for your careful review. We appreciate your encouraging comments about the writing, soundness, scope, and applicability. We have responded to your suggested areas for improvement below.
>
> **Originality: ... On the other hand, there is a highly relevant line of prior work that is not cited or discussed: "AIR” ... A detailed discussion of how the proposed approach relates to this prior work, and potentially comparison against it in the evaluation is in my opinion necessary.**
>
> Thank you for this very helpful comment. We will include a citation of the Eslami et al. paper and a discussion of how it relates to our work within the "Bayesian inverse graphics" section of the Related Work section.
>
> In response to your comment, we investigated qualitatively how a version of our approach that was modified to use an RGB likelihood instead of a depth likelihood performs on synthetic RGB images of scenes that resemble those used by Eslami et al. Our method uses no deep learning initialization and assumes knowledge of an accurate 3D model of object types (just as in Eslami et al.) The [results](https://bit.ly/3m45VGb) show that our method produces parses that are qualitatively more accurate than the method of Eslami et al. We expect this is due to a combination of factors, including use of blur in their RGB likelihood and downsampling image inputs to the inference network, and our use of iterative model-based inference as opposed to amortized inference. We also note the method of Eslami et al. took 3 days to train, whereas our method did not require any training.
>
> We view the approach of Eslami et al. as complementary to ours, and integrating the two approaches is interesting grounds for future work: It seems potentially useful to employ an inference model based on a deep neural network that takes as input an RGB-D frame and samples a scene graph, using a stochastic choice at each iteration to decide whether or not to continue growing the graph. This generalizes the architecture of Eslami et al., which can be seen as proposing from a restricted set of scene graphs in which all objects have an edge connected to a node for the table. Such an inference model could be trained either via mode-seeking loss (as in Eslami et al.) or via supervision from synthetic data simulated from the generative model, and could be combined with our Monte Carlo framework in several ways (via initialization, importance sampling, or to produce proposals within an MCMC framework) to potentially achieve scene parsing that is as accurate as our current results but faster. In our current experiments, we use DenseFusion to initialize each object pose independently, but using an inference model (as proposed by Eslami et al.) that proposes to object instance presence and pose variables jointly is one natural next step.
>
> We do not think Eslami et al.'s AIR approach is a viable baseline. It does not apply to the standard YCB-Video 6DoF pose benchmark task, because of its restrictive assumptions about object contact and its 3DoF (rather than 6DoF) pose representation. Second, AIR does not provide quantitative results on real 3D scenes --- only simple synthetic scenes. Third, there do not appear to be any public implementations of AIR for 3D scenes (only for 2D images) --- nor sufficient detail to replicate the approach they used for simultaneous domain randomization and stable gradient estimation. Finally, they describe challenges in applying their method to real 3D scenes (“robust transfer [to real data] remains a challenging problem in general") which would make it difficult to compare on our real datasets.
>
>
> **Quality: … there is no evaluation of the object shape priors either independently, or as part of the proposed approach. In other words, the impact of Section 4 on the results is not clear in the current paper.**
>
> In all experiments discussed in Section 5, the object shape priors are learned from 5 depth images using the procedure described in Section 4. Our inference algorithm does not have access to the ground truth shape models, only those learned by this procedure. Thus the quantitative evaluation done in Section 5 implicitly tests the quality of the shape-prior learning. If the learned shape priors did not match the true shape model well, it would make accurate pose estimation difficult. Our results show high accuracy pose estimation (improving over baselines), showing that our shape prior learning is sufficiently accurate for use on real datasets. Also see these [results](https://bit.ly/3CDlryg) showing shape learning from real data.
>
> In response to your comment, we have also performed a [quantitative evaluation comparing the learned shape models to the ground truth shape models](https://bit.ly/3CDlryg). To get a shape model from the learned shape prior, we take all voxels which the prior says are more likely to be occupied than unoccupied and compute the IoU between that volume and the ground truth object volume.
>
> We will also include 1) quantitative comparisons to simpler geometric shape estimation methods (e.g. bounding boxes, convex hulls) that have been used in robotics tasks, and 2) pose estimation results using these less accurate shape models, showing that pose estimation degrades substantially with inaccurate shapes (we have seen this qualitatively).
>
> In response to your comments, we also verified the [quantitative accuracy of SLAM for shape acquisition](https://bit.ly/3jH8xqr). We report the mean translation and rotational errors of pose with respect to the true poses from which the 5 depth images were generated. These errors (under 2 centimeters, and roughly 1 degree of rotational error) are consistent with both the high accuracy shape models we learned, and also the rough order-of-magnitude quantitative errors that we observed in pose estimation on real data.
>
> Finally, we also want to highlight that while we have presented an approach to learning shape priors, our framework is sufficiently general to support many methods of specifying these object shape prior distributions e.g. we could hand-construct the prior distribution or train a GAN given a dataset of object instances. In future work, we plan to develop our shape learning further and investigate how we can include constraints on convexity, continuity, etc. and general prior knowledge about shape to learn more realistic shape priors and demonstrate we can learn from images with objects in clutter or in-hand and ideally from just a single image.
>
>
> **Significance: … However, the empirical results only partially back up the complexity of the proposed approach, and some discussion of relation to prior work is missing (see above points). Therefore, I am not as positive as I could be with respect to acceptance.**
>
> Thank you for your encouraging feedback about scope and generality. We hope the additional empirical results for shape learning, the discussion of the relationship to Eslami et al., and the qualitative results showing that our approach is more accurate in pose estimation help to address these concerns.
>
> **Some limitations are discussed at a high level in the last section.  … Potential negative societal impacts were not discussed**
>
> We will prominently discuss limitations around speed in the final version of the paper.
> Speed is the main limitation of our approach. See our response to Reviewer 3 for a more detailed discussion of speed, including quantitative data for scenes with varying numbers of objects, that we will prominently include in the final paper.
> We see two clear directions for improving this: (1) perform the likelihood computation on a GPU and batch rendering and (2) a coarse-to-fine inference algorithm that varies the resolution (of observation and latent variables) over the course of inference (initially operating on lower resolution data which can be fast, and then increasing the resolutions to improve accuracy of estimates). We are excited to evaluate gains from coarse-to-fine inference (so more inference is done on faster, lower-resolution versions), ordinary software performance engineering (to speed up likelihoods), and also graph neural net proposals (to reduce the amount of MCMC, for fixed accuracy) could together close the speed gap with DenseFusion.
>
> We have added the following Societal Impact section at the end of the paper:
>
> 3D scene graph perception is potentially useful as a central computer vision component in diverse application domains such as AR, security/surveillance, HCI, and robotics. This paper shows how to improve robustness, accuracy, and learnability/data efficiency, relative to pure deep learning approaches. These improvements could make it easier to build and maintain beneficial systems as well as harmful ones. Examples of potentially beneficial applications include assistive indoor AR/mixed-reality for the cognitively impaired; some indoor safety/security applications; and educational technology. Examples of potentially harmful applications include widespread, automated analysis of humans and their interactions with the physical environment, by nation states and for-profit corporations, and the development of improved indoor perception & target tracking systems for autonomous weapons.
>
> This paper also opens up new avenues for research, as well as new potential risks, in dataset bias and disparate impact for 3D perception, via its use of human-editable probabilistic programs as scene graph priors.

---

> > ### Author Response · Authors · 2021-09-08
> > **Checking In**
> >
> > We are checking in again to see if there is any feedback/comments in response to the additional experiments and clarifications we have provided. We appreciate the feedback you have provided thus far and want to ensure that we have properly addressed all your concerns!

---

### Official Review · Reviewer_6dXH · 2021-07-19

**Rating:** 6
**Confidence:** 2

**Summary:**

This paper presents a method for multi-object 6DoF pose estimation with a "probabilistic programming" model. Before inference, the model learns priors on 3D occupancy/shape for objects. Then, given the number of objects in the scene and the classes for objects in a test scene (observed in a depth map), inference proceeds by MCMC sampling scene graphs and poses and occupancies, to minimize a depth reconstruction error. The results look good on YCB and on a new synthetic dataset based on YCB objects, against two recent 6DoF pose estimators, showing good results.

**Ethical Concerns:**

No concerns.


**Limitations And Societal Impact:**

No concerns.


**Main Review:**

This paper was very hard for me to understand, and there still may be parts I have misunderstood. I will set up my review as a list of confusion points, which might all be easily answered by the authors.

- What is the purpose of assigning the names top/bottom/left/right/front/back to the planes of the box? It doesn't seem like this information is ever used.

- How exactly do you model contact relationships, and learn priors about them? I found many variables that relate to contacts in the "Scene graphs" paragraph (89-109) but they never come up again, either in the generative model or the inference algorithm. In general there is a lot of notation for the reader to keep track of, and it seemed like much of it was declared and then forgotten, so perhaps it can just be deleted. (But I do very much want to learn about those contact relationships.)

- The voxel grids have "fixed dimensions h, w, l ∈ N", but they also have a "grid resolution s ∈ R". What is the difference between a dimension and a resolution?

- The text says that the scene graph prior (equation 1) "favors scenes that can be parsimoniously described via either fewer objects, or fewer objects with independently-sampled 6DoF poses". The idea of "fewer objects" occurs in both sides of the "either", so it seems that part should always be true, but actually the number of objects is held constant by an assumption stated earlier: "an a-priori known number of objects N0 (used in the experiments in Section 5)". So the parsimonious/fewer part seems inactive here, and we are just left with "independently-sampled 6DoF poses". It seems like little-to-no information or inductive bias has actually been introduced here. Am I reading that right?

- Is the pre-trained object occupancy prior conditioned on class?

- The paper says the model performs well "despite being trained on only a few synthetic depth images". How many is a few? (Is it the "5 depth images of each object type" mentioned later? Why does the abstract say "5 or fewer"?) Also, why was it trained on only a few? Since they are synthetic, I expect them to be plentiful.

- Do the baselines also assume known class and number of objects? They seem to involve detection and segmentation as well as pose estimation, so maybe they are a harder problem, and it's not an apples-to-apples comparison.

- I do not quite follow Table 1, and how to compare it with prior work. The Robust6D and DenseFusion papers report ADD-S accuracy across far more classes (~21 instead of 5), their values are in the range 90-100, and they also report the mean across all classes. Table 1 reports "ADD-S error" at three quartiles, and the values are under 2, and it looks like the mean across objects is never calculated. It would be good to do the same kind of evaluation as the previous work.

**Time Spent Reviewing:**

3

---

> ### Author Response · Authors · 2021-08-10
> **Author Response**
>
> Thank you for all of your detailed questions. Making these elements clearer will surely improve our paper.
>
> **The paper says the model performs well "despite being trained on only a few synthetic depth images"...**
>
> Thank you for these questions. The deep learning baselines were trained using thousands of synthetic images generated using high-quality object models. In contrast, we learned our shape priors for each object from only 5 synthetic depth images, highlighting the data-efficiency of our pipeline compared to the baselines. The "or fewer" language is an artifact of earlier experiments which used fewer than 5 images but these were not included in the paper. We will remove “or fewer” from the abstract. Additionally, while the gap between synthetic and real depth data is much smaller than for RGB data, to demonstrate that the object-learning method in Section 4 also works on real depth images, we have included a [demonstration](https://bit.ly/3Aw6NqK) of a shape model learning from 5 real depth images of a mug object. We will clarify these items in the main text and discuss data-efficiency in more detail, as it is an important advantage we see in our approach.
>
> **Do the baselines also assume known class and number of objects? .. and it's not an apples-to-apples comparison.**
>
> This is a valuable question and we realize it is important to clarify the evaluation protocols. We follow the evaluation approach of the standard well-known BOP challenge [1]. The BOP challenge evaluates accuracy of pose estimates when the provided inputs are (1) an RGBD image and (2) the number of each object type that is present in the image (See [BOP Task Definition](https://bop.felk.cvut.cz/challenges/)). This setup avoids confounding pose estimation with object detection. We assume these same inputs for our evaluation. To ensure an apples-to-apples comparison with DenseFusion and Robust6D, for each input image, we check that the baseline's internal object detector outputs the correct number of each object type. We only evaluate on images for which the baselines did output the correct number of each object type. We will clarify this in text to make it clear that the comparison is apples-to-apples.
>
> To be clear, 3DP3 _does apply_ to the more general setting where just an RGBD image is given as input.  3DP3 can incorporate object detectors and pose estimators as components, and use the object detectors to set the number and types of objects before MCMC inference. 3DP3 supports both DenseFusion and also the kinds of detectors described in Supplemental Section F (that do not require any labeled training data, and instead can be constructed from just the 5 real images that 3DP3 uses to learn models of objects). 3DP3 then improves the robustness and accuracy of the pose inferences. 3DP3 can be run using numbers of object types that are output from object detectors --- or from a side channel, as in the BOP evaluation protocol.
>
> Given the definition of the BOP protocol it is functionally equivalent to (i) run the baseline detector as an input to 3DP3 (to set the number of objects of each type --- which is restricted to scenes where this set of detections is the same, for all methods, by definition of the protocol) as it is to (ii) run 3DP3 given the ground-truth number of objects of each type, and to only use detectors for bottom-up pose proposals (or to forego bottom-up detectors entirely, during pose estimation). In fact, on YCB-Challenging scenes, we don't use any bottom-up detectors for pose estimation.
>
> We think an important future research direction to explore is using 3DP3 to improve object detection, e.g. by pruning false detections, or by doing a large amount of top-down MCMC to detect objects that all bottom-up detectors missed. However, we have not performed a quantitative evaluation of this kind of hybrid approach in this paper, and restrict our claims to pose estimation, following the BOP protocol.
>
> [1] Hodaň et al. ECCV 2020
>
>
> **The text says that the scene graph prior (equation 1) "favors scenes that can be parsimoniously described via either fewer objects, or fewer objects with independently-sampled 6DoF poses" ...**
>
> We apologize for the confusing wording.
>
> The second clause was intended to be parsed as "fewer (objects with independently-sampled 6DoF poses)", that is, even among scenes with some fixed number $N_0$ of objects, we favor those scenes in which fewer of the objects have independently sampled 6DoF poses i.e. more object poses are explained by contact relationships.  We intended to make two general points about how the posterior distribution induced by our class of generative models utilizes Bayesian Occam's razor to adjust the complexity of the scene graph explanations as needed to explain a given point cloud. We describe more precisely below:
>
> Suppose that we observe a point cloud that is fit well by a scene graph with a single object ($N = 1$). The marginal likelihood of that point cloud given a scene graph structure where $N = 1$ will be higher than the marginal likelihood given a scene graph structure where $N = 2$ because the probability that two objects are approximately aligned with the point cloud is lower than the probability that one object is aligned. In this sense the model will favor explanations that are more parsimonious because they use fewer objects (and you are correct to say this is not relevant for our experiments in Section 5).
>
> Similarly, suppose there exists some scenegraph with one object (e.g. a box) in planar contact with another object (e.g. the table), fitting a point cloud well. The marginal likelihood of the point cloud given that $N = 2$ and an edge exists between the two nodes will be higher than the marginal likelihood if $N = 2$ and no edge because the probability mass assigned to scenarios where the two objects are in approximate planar contact is much less when there is no edge in the graph. In this sense, for fixed $N$, the posterior will prefer explanations that utilize fewer independent 6DoF poses and will prefer explanations where the objects are in planar contact.
>
>
> **I do not quite follow Table 1, and how to compare it with prior work. ...**
>
> Table 1 presents results of an evaluation on our synthetic YCB-Challenging dataset which only features 5 of the 21 objects. Our evaluation on the YCB-Video dataset containing all 21 objects is in the Supplementary Material Table 2, in which we perform the same kind of evaluation as is done in the DenseFusion and Robust6D papers. We will include the mean across classes in that table. Based on this feedback, we think it makes sense to bring Supplementary Table 2 into the main text, as it is the primary evaluation that conveys our approach outperforms baselines on an existing standard dataset.
>
> Regarding the ranges of values, in Table 1 we are reporting a mean ADD-S error which is in the range of 0-2cm. Supplementary Table 2, reports accuracy at a threshold on ADD-S which is in the 90-100 range. We will clarify the difference.
>
>
> **The voxel grids have "fixed dimensions $h, w, l \in \mathbb{N}$, but they also have a "grid resolution $s \in \mathbb{R}$. What is the difference between a dimension and a resolution?**
>
> We will clarify this in the revision.
>
> The voxel grid is composed of many voxel cells. The resolution $s$ defines the size of each voxel cell (an $s \times s \times s$ cube).  The dimensions $h$, $w$, $\ell$ define the size of the voxel grid in terms of how many voxel cells high, wide, and long it is. This means the actual size of the voxel grid is $hs \times ws \times \ell s$.
>
>
> **What is the purpose of assigning the names top/bottom/left/right/front/back to the planes of the box?**
>
> These names are just for human readability. The algorithm maintains numerical indexes, for the bounding boxes of all learned objects.
>
> In our generative model for objects in contact we sample which faces of the parent and child 3D bounding box are in contact. The choices of the specific faces (along with the other sampled contact parameters) define the relative pose between the parent and child object. For example, we might sample a contact between a table object and a mug object such that the top of the table is in contact with the top of the mug. In that case, the pose of the mug would be set such that the mug is upside down on the table surface. We chose the names Top, Bottom, Left, Right, Front, Back for these faces rather than e.g. 1, 2, 3, 4, 5, and 6 so their meaning would be clearer to the reader. We will add more annotations to Figure 1 to make it clearer what the contact parameters are and how they together define a relative pose.
>
>
> **How exactly do you model contact relationships, and learn priors about them? I found many variables that relate to contacts in the "Scene graphs" paragraph (89-109) but they never come up again, either in the generative model or the inference algorithm.**
>
> We apologize for the dense notation. We will add more details to Figure 1, and also a table that defines notation and links it to where it is introduced in more detail.
>
> The parameters of a contact relationship (lines 101-109) define a relative pose between the contacting object and its parent. The generative model places prior distributions on each of these parameters (lines 132-135). For our experiments we used broad prior distributions, automatically constructed from the object shape models without tuning to a specific scene, which encourage near-flush planar contact. It is possible to learn more informative priors on contacts from data (e.g. learn which face of an object is more likely to be in contact with which face of another object, or what region on a table surface an object is likely to be in contact with), but these extensions were not necessary for the pose accuracy improvements shown in our experiments.
>
> **Is the pre-trained object occupancy prior conditioned on class?**
>
> Yes, the shape prior is conditioned on object class.

---

> > ### Comment · Reviewer_6dXH · 2021-09-01
> > **OK**
> >
> > Thank you for these answers. I think things are getting a bit clearer now, but I still have many questions.
> >
> > Instead of the google doc with additional experiments, I would have much preferred a new figure or two, with more detailed notation, as the rebuttal itself suggests.
> >
> > > The second clause was intended to be parsed as "fewer (objects with independently-sampled 6DoF poses)"
> >
> > This new quote does not seem to fit anywhere in the sentence we are talking about. Can you just write the whole new sentence? The two paragraphs you wrote following this quote made some sense.
> >
> > I am still a bit worried about the scope and clarity of the method description, which reviewer NB9L pointed out also. As I quoted earlier, the text assumes "an a-priori known number of objects N", but still we have this confusing discussion about how "fewer objects" plays a part, and the text discusses binomial and poisson distributions for the prior on number of objects, and later mentions sampling from these distributions -- yet all of this seems completely irrelevant to what is really happening in the real model, where N is assumed given by an oracle. I know that the supplemental has more material and more extensions with more relaxations and so on, but my priority is for the method in the main paper to make sense.
> >
> > > We only evaluate on images for which the baselines did output the correct number of each object type.
> >
> > This does not seem sufficient, because your model assumes known number *and known class*. The baselines do not have this privileged information.
> >
> > > To be clear, 3DP3 does apply to the more general setting where just an RGBD image is given as input. 3DP3 can incorporate object detectors and pose estimators as components, and use the object detectors to set the number and types of objects before MCMC inference.
> >
> > Doing a task with your method, vs doing a task with a different method, are not the same thing. It seems to me that 3DP3 **does not** apply to the more general setting where just an RGBD image is given as input, since as you mention, it involves running other methods as "components".
> >
> > Probing a little deeper into a "negative" interpretation (partly to help me understand a little better), I would like to ask: could it be that the handcrafted priors defined in this model are only applicable to YCB and similar datasets, and would actually make correct solutions practically impossible to return when the assumptions are gently violated? The prior distributions do not seem particularly "broad" as suggested here. For example, imagine one object leaning on another, so that no two faces are in flush contact. Then, the carefully chosen prior $z_v \sim N(0, 1cm)$ (the perpendicular distance between two faces) will prevent a good solution from being found, since the true face distances will be in an extremely low-probability region of the distribution. So, the method is maybe only useful for objects that are either sitting upright or lying flat down (among other assumptions). Is that fair to say?
> >
> > I am confused about the parent-child relationships between objects, in Figure 1 and in general. Starting with Figure 1, why is the soup can the child of the spam can? When, in general, does an object-object parent-child relationship happen during inference? It's clear that the parent defines the coordinate system for the child, but it's not clear why this helps with inference. My one guess is that this has to do with the flushness prior: if one object is sitting flat on top of another, then a parent-child relationship there will fit.
> >
> > How does the model deal with intersections? (How do you prevent the model from inferring a scene where two objects intersect?)

---

> > > ### Author Response · Authors · 2021-09-02
> > > **Author Response (Part 1)**
> > >
> > > **Instead of the google doc with additional experiments, I would have much preferred a new figure or two, with more detailed notation, as the rebuttal itself suggests.**
> > >
> > > We will work on creating these new figures, add them to the google doc [here](https://docs.google.com/document/d/1Y44ZmDCEE20WadRVf7q4IyX2bgMKvKLChll9ysnQAgA/edit#bookmark=id.igndinpatuns), and follow up with you.
> > >
> > > **> The second clause was intended to be parsed as "fewer (objects with independently-sampled 6DoF poses)"**
> > >
> > > **This new quote does not seem to fit anywhere in the sentence we are talking about. Can you just write the whole new sentence? The two paragraphs you wrote following this quote made some sense.**
> > >
> > > **I am still a bit worried about the scope and clarity of the method description, which reviewer NB9L pointed out also. As I quoted earlier, the text assumes "an a-priori known number of objects N", but still we have this confusing discussion about how "fewer objects" plays a part, and the text discusses binomial and poisson distributions for the prior on number of objects, and later mentions sampling from these distributions -- yet all of this seems completely irrelevant to what is really happening in the real model, where N is assumed given by an oracle. I know that the supplemental has more material and more extensions with more relaxations and so on, but my priority is for the method in the main paper to make sense.**
> > >
> > > We apologize for the continued confusion. We will change our approach to conveying this intuition on how the scene graph structure posterior behaves.
> > >
> > > First, we will modify the "Prior on scene graphs" paragraph (lines 118 - 139) in Section 3.1 and Figure 2 to only describe the case when $N$ is assumed known. We will move the extensions to the case of unknown $N$ (including use of binomial and Poisson distributions) into Section G of the supplement which has experiments demonstrating how $N$ can be inferred using Bayesian inference.
> > >
> > > Second, we will remove the confusing sentence on lines 137-139 altogether, and replace it with a new paragraph in Section 3.2 after line 186, where the MCMC kernel over scene graph structure is described. The new paragraph will be:
> > >
> > > "To understand the role of this MCMC kernel, consider the case when there are just two objects: a table ($v_1$) and a box ($v_2$), so that $V = {r, v_1, v_2}$ (recall that $r$ is the root node representing the world coordinate frame). Then there are three possible scene graph structures $G = (V, E)$ for these two objects that differ only in what edges are present in the graph: $E_1 = {(r, v_1), (r, v_2)}$, $E_2 = {(r, v_1), (v_1, v_2)}$, and $E_3 = {(r, v_2), (v_2, v_1)}$. Suppose we observe a point cloud in which the box is standing upright and its bottom face is in flush contact with the table, and suppose the current state of the Markov chain has scene graph structure $G_1 = (V, E_1)$ (the two objects are not connected by an edge) and scene graph parameters such that both objects have correct 6DoF poses (i.e. the rendered point clouds line up perfectly with the observed point cloud). Suppose the kernel proposes to change the scene graph structure to $G_2 = (V, E_2)$ (so that there is an edge from the table to the box). This proposal has high probability of being accepted, since the prior on scene graph parameters $p(\theta | G_2)$ puts more probability mass on 6DoF object poses that are near the correct 6DoF poses (because it concentrates its probability mass on poses in which the objects are nearly in flush contact) than does $p(\theta | G_1)$ (which distributes its probability mass amongst many poses that are far from flush contact)."
> > >
> > > We could keep the sentence on lines 137-139 and remove the “either fewer objects or” part, but we now think it is probably better to remove this sentence altogether, since the longer paragraph above makes the same point, but with more concrete detail and with sufficient context regarding scene graph structure inference.
> > >
> > > **> We only evaluate on images for which the baselines did output the correct number of each object type.**
> > >
> > > **This does not seem sufficient, because your model assumes known number and known class. The baselines do not have this privileged information.**
> > >
> > > We are providing exactly the same information to the baselines as is provided to our model. When we say “We only evaluate on images for which the baselines did output the correct number of each object type.”, this means that we are giving the baseline _both_ the number and the classes of each object. (For a scene with 2 apples, 1 ball, and 3 chairs, we verify that the baseline made exactly 2 pose estimates for apples, 1 pose estimate for the ball, and 3 pose estimates for chairs.)
> > >
> > > **> To be clear, 3DP3 does apply to the more general setting where just an RGBD image is given as input. 3DP3 can incorporate object detectors and pose estimators as components, and use the object detectors to set the number and types of objects before MCMC inference.**
> > >
> > > **Doing a task with your method, vs doing a task with a different method, are not the same thing. It seems to me that 3DP3 does not apply to the more general setting where just an RGBD image is given as input, since as you mention, it involves running other methods as "components".**
> > >
> > > You are absolutely correct that Section 5 evaluates pose estimation accuracy by providing side information beyond the RGBD image, specifically the number of objects of each type. We also agree with you that it is clearer to leave Bayesian inference of $N$ and $\mathbf{c}$ via MCMC to an appendix, and focus on clearly explaining the variant of 3DP3 inference that assumes $N$ and $\mathbf{c}$ are given before MCMC begins. Quantitatively evaluating 3DP3 with Bayesian inference over $N$ and $\mathbf{c}$ raises additional challenges that we do not address in this paper.
> > >
> > > We regret that our description of 3DP3 did not clearly explain exactly how 3DP3 uses object detectors (instead only briefly alluding to it on line 167), and are grateful for help in clarifying this aspect of the paper!
> > > Here is how 3DP3 works on RGBD images, without Bayesian inference over $N$ and $\mathbf{c}$. The first step is to run bottom-up object detectors over the image --- either our own bottom-up detectors (from Supplemental Section F), or the detection stage from a deep learning method such as DenseFusion. The number of detections determines $N$, and the detected object classes determine $\mathbf{c}$, together specifying the scene graph prior with fixed $N$ and $\mathbf{c}$. Initial pose estimates can be provided either by bottom-up neural pose estimators, or via the ICP-based procedure (described in Supplement Section F). Given $N$, $\mathbf{c}$, and initial pose estimates, 3DP3 then performs scene graph structure and parameter inference via MCMC. For the YCB-Video dataset, we used DenseFusion as the object detector in 3DP3, and we also used DenseFusion for initial pose estimation.
> > >
> > > Our final paper will include pseudocode for this full algorithm.
> > >
> > > The widely-used BOP evaluation protocol for 6DOF pose estimation restricts evaluation of pose estimation to scenes with correct $N$ and $\mathbf{c}$, for *both* 3DP3 and DenseFusion --- i.e. only those scenes where DenseFusion detected the correct number of objects of each class. The pose estimation results we report thus constitute an "apples-to-apples" comparison between 3DP3 (using DenseFusion detection & pose initialization) and DenseFusion on its own. While we could have evaluated both object detection and pose estimation accuracy, our pose estimation method uses the same object detection component as the baselines, so the additional results would not have been informative.
> > >
> > > We thank you for your careful reading that surfaced this lack of clarity in our writeup!

---

> > > > ### Comment · Reviewer_6dXH · 2021-09-02
> > > > **OK**
> > > >
> > > > This is making a lot more sense now. Please write the paper this way! And even for the paragraphs you have committed to replacing, please continue improving the clarity, because I am sure it can be better still.
> > > > I will increase my score now. Best of luck.

---

> > > ### Author Response · Authors · 2021-09-02
> > > **Author Response (Part 2)**
> > >
> > > **Probing a little deeper into a "negative" interpretation (partly to help me understand a little better), I would like to ask: could it be that the handcrafted priors defined in this model are only applicable to YCB and similar datasets, and would actually make correct solutions practically impossible to return when the assumptions are gently violated? The prior distributions do not seem particularly "broad" as suggested here. For example, imagine one object leaning on another, so that no two faces are in flush contact. Then, the carefully chosen prior
> > > zv∼N(0,1cm)
> > >  (the perpendicular distance between two faces) will prevent a good solution from being found, since the true face distances will be in an extremely low-probability region of the distribution. So, the method is maybe only useful for objects that are either sitting upright or lying flat down (among other assumptions). Is that fair to say?**
> > >
> > > Thank you for this good question and the opportunity to clarify.
> > >
> > > Our method can correctly infer object poses even for scenarios like you describe. The YCB-Video test set contains many images of a scene where an Expo marker is leaning, at an angle, on a wooden block and still, 3DP3 is able to correctly infer its pose. (We have included include [examples of such images](https://docs.google.com/document/d/1Y44ZmDCEE20WadRVf7q4IyX2bgMKvKLChll9ysnQAgA/edit#bookmark=id.ystrudgzu0ot) in the google doc)
> > >
> > >
> > > This is possible because our scene graph prior distributes probability mass over scene structures in which objects have independent 6DoF poses as well as scene structures in which objects are in flush contact with one another. For example, our scene prior always places some prior probability on the graph structure (denoted $G_0$ on line 128) where every object is a child of the root node, and therefore has its full 6DoF pose sampled independently from a broad prior (uniform 3D position over the spatial extent of the scene and uniform 3D orientation). Because we jointly infer both the parameters and the structure of the scene, the model is able to automatically determine whether the "flush object-object contact" assumption should be utilized or not given the observations, via Bayesian inference over scene structure.
> > >
> > > Including the possibility of flush contact in our model is an inductive bias: On a test data set where all objects were floating in space at independently sampled 6DoF poses, our method would likely perform less well than the ablated version of our method where the scene graph structure was fixed to $G_0$ (no edges between objects in the graph). However, on a test data set where objects are often (although not always) in flush face-to-face contact with the tabletop or other objects, adding this inductive bias gives more accurate fine-grained pose estimation relative to the ablation (see Figure 6 and Supplement Table 2).
> > >
> > > The numerical details of our prior were also not tuned or tailored in any detailed way: We use a uniform distribution over all scene graph structures over $N$ objects. The 1cm standard deviation for perpendicular face-to-face distance, the 50cm standard deviation for in-plane face-to-face offset were not tuned to the benchmark data sets -- they were chosen heuristically based on the rough dimensions of table-top objects. The volume of the space, and the concentration parameter $\kappa$ were also not tuned to the benchmark data sets. We chose these simple, generic priors to show the robustness and ease of use of our approach.
> > >
> > > **I am confused about the parent-child relationships between objects, in Figure 1 and in general. Starting with Figure 1, why is the soup can the child of the spam can? When, in general, does an object-object parent-child relationship happen during inference? It's clear that the parent defines the coordinate system for the child, but it's not clear why this helps with inference. My one guess is that this has to do with the flushness prior: if one object is sitting flat on top of another, then a parent-child relationship there will fit.**
> > >
> > > The soup can is a child object of the spam can because in the scene, the soup can is sitting on top of the spam can. If this is unclear, we can regenerate the image from a different viewpoint which makes it more obvious.
> > >
> > > We hope our response to the previous comment helps to clarify why object-object parent-child relationships help with inference. To put it another way, when one object is sitting flat on top of another, then posterior inference will infer that there is probably an object-object edge, and it will then encourage the 6DoF poses of the two objects to be more closely aligned with one another than if object-object contact relationships were not modeled. In effect, object-object edges allow the point cloud data for one object to inform the pose of another object, and vice versa; without object-object edges, an object's pose is only informed by its own part of the point cloud.
> > >
> > > **How does the model deal with intersections? (How do you prevent the model from inferring a scene where two objects intersect?)**
> > >
> > > Our model does not explicitly prevent inferring scenes with intersecting objects. Our scene prior, specifically the prior over contact parameters, encourages explanations with approximately flush contact. (The prior over the direction alignment of the child object concentrates on perfectly flush contact. And the prior over the offset distances between the child and parent object contact planes is concentrated at 0, perfect alignment). As a result, the inference algorithm often converges to explanations with no intersections. Also in most scenes, the observed data would not support such an explanation of objects intersecting one another.
> > >
> > > Note that it is possible to extend our pose estimation algorithm to penalize object interpenetration more generally, by introducing a soft constraint as an additional likelihood term  that penalizes the total amount of interpenetration. This may improve our pose estimation results, but we did not find this necessary to outperform the baselines.

---

### Author Response · Authors · 2021-08-10
**Summary of Author Response**

We thank the reviewers for their very detailed and careful reviews, which have helped us improve the paper. In response to your questions and comments, we have added three experimental results:

(1) a [quantitative evaluation of our shape model learning](https://bit.ly/3CDlryg)

(2) a [demonstration of our shape-learning module](https://bit.ly/3Aw6NqK) running on 5 real depth images (instead of on synthetic depth images)

(3) a [qualitative comparison to Attend, Infer, Repeat by Eslami et al.](https://bit.ly/3m45VGb) using a modified version of our model that parses RGB images rather than depth images

The final version of the paper will include these results. We will also prominently explain how our empirical comparison is based on the well-known BOP benchmark (yielding an apples-to-apples comparison between our method 3DP3 and the deep learning baselines), and also more prominently quantify the speed / accuracy tradeoffs of 3DP3.

---

### Author Response · Authors · 2021-08-24
**Checking In**

We are checking in to see if there are any additional experimental results or clarification that we can provide. The feedback we have received from each of you is greatly appreciated and has already helped to improve our paper. If you have further comments, thoughts, or concerns we are happy to engage and discuss.

---

### Decision · Program_Chairs · 2021-09-27

**Decision:**

Accept (Poster)

**Comment:**

This paper presents a method for multi-object 6DoF pose estimation with a "probabilistic programming" model. Before inference, the model learns priors on 3D occupancy/shape for objects. Then, given the number of objects in the scene and the classes for objects in a test scene (observed in a depth map), inference proceeds by MCMC sampling scene graphs and poses and occupancies, to minimize a depth reconstruction error.
The reviewers raised concerns regarding fairness of comparisons to baselines, known versus unknown number of objects at inference time, violation of priors, (lack of) comparisons to other object-centric generative models. The rebuttal of the authors addressed those concerns and generated extensive discussion. The paper is suggested for publication.